# MAGIC-Flow: Multiscale Adaptive Conditional Flows for Generation and Interpretable Classification

## Abstract

Generative modeling has emerged as a powerful paradigm for representation learning, but its direct applicability to challenging fields like medical imaging remains limited: mere generation, without task alignment, fails to provide a robust foundation for clinical use. We propose MAGIC-Flow, a conditional multiscale normalizing flow architecture that performs generation and classification within a single modular framework. The model is built as a hierarchy of invertible and differentiable bijections, where the Jacobian determinant factorizes across sub-transformations. We show how this ensures exact likelihood computation and stable optimization, while invertibility enables explicit visualization of sample likelihoods, providing an interpretable lens into the model's reasoning. By conditioning on class labels, MAGIC-Flow supports controllable sample synthesis and sound class-probability estimation, effectively aiding both generative and discriminative objectives. We evaluate MAGIC-Flow against top baselines using metrics for similarity, fidelity, and diversity. Across multiple datasets, it addresses image classification and generation across different scanners, modality-specific image synthesis, and the classification of patient images with diverse clinical diagnoses. Results show MAGIC-Flow creates realistic, diverse samples and improves classification. MAGIC-Flow is an effective strategy for generation and classification in data-limited domains, with direct benefits for privacy-preserving augmentation, robust generalization, and trustworthy medical AI.

## 1 Introduction

Generative modeling has become a cornerstone of modern machine learning, powering advances in representation learning, data augmentation, and controllable synthesis. In addition to semi-supervised and transfer learning, generative models have been increasingly used to mitigate data scarcity. GANs (Goodfellow et al., 2014; Makhlouf et al., 2023; Han et al., 2018), VAEs (Sohn et al., 2015; Diamantis et al., 2022), and diffusion models (Nichol & Dhariwal, 2021; Dhariwal & Nichol, 2021; Hung et al., 2023; Kazerouni et al., 2022) have been proven effective at synthesizing realistic samples and improving classifier robustness (Shorten & Khoshgoftaar, 2019; Shin et al., 2018). Yet, in safety-critical domains such as medical imaging, their limitations are clear. Datasets are small, expensive to curate, and biased by acquisition protocols (Zech et al., 2018). Models that generate realistic images without offering *task alignment* or *interpretability* fall short in clinical contexts, where reliability and transparency are critical. Moreover, adversarial and diffusion-based models suffer from mode collapse, hallucination, and instability (Arjovsky et al., 2017; Cohen et al., 2018; Yi et al., 2019), undermining their trustworthiness.

Normalizing flows provide a principled alternative. Because flows enable exact *likelihood estimation*, *invertible mappings*, and *stable, non-adversarial training*, they are well suited to data-limited, safety-critical use cases. However, existing architectures such as RealNVP (Dinh et al., 2016) and Glow (Kingma & Dhariwal, 2018) were designed primarily for *generation* and typically handle classification by attaching auxiliary discriminative heads or training downstream classifiers on latent embeddings. This decoupled design prevents a unified treatment of generation and classification and leaves likelihoods under-utilized for decision-making. Furthermore, current flow variants—such as Flow++, cINNs, CaFlow, and MARFlow—do not jointly address the stringent constraints of

medical imaging: limited labels, high-resolution structure, fine-grained class separability, and strict interpretability requirements.

In this context, we introduce **MAGIC-Flow**, a conditional multiscale normalizing flow that unifies *generation and classification* within a shared invertible architecture. Unlike canonical flows, which remain purely generative, and unlike hybrid approaches that attach external discriminators, MAGIC-Flow is built around a *novel coupling transformation* and a *multi-task, fully invertible pipeline* that enables: (1) expressive, label-conditioned sample generation, (2) the extraction of class-conditional likelihoods for transparent, quantitative classification, (3) a multi-scale factorization optimized for high-resolution medical images, and (4) a discriminative pathway grounded entirely in invertibility rather than opaque latent embeddings. This unified design is motivated by the unique challenges of medical imaging, where models must operate under extreme data scarcity, preserve fine anatomical structure, remain stable at small batch sizes, and provide interpretable uncertainty estimates. MAGIC-Flow directly addresses all of these constraints—none of which are jointly handled by existing generative or flow-based architectures.

We evaluate MAGIC-Flow on three clinically relevant problems: (i) classification and generation under scanner noise, where classes are defined by acquisition artifacts rather than semantic content; (ii) cross-modality generation across imaging protocols; and (iii) diagnosis classification, where the model must discriminate between clinically meaningful disease categories based on fine-grained anatomical and physiological differences. Across all settings, MAGIC-Flow consistently improves over state-of-the-art generative and discriminative baselines in terms of *sample fidelity, diversity, classification accuracy, and likelihood-based interpretability*, demonstrating its effectiveness as a trustworthy framework for medical AI.

**Contributions.** Our main contributions are threefold:

- We propose MAGIC-Flow, the first conditional multiscale normalizing flow that supports both generation and classification using the same invertible backbone and a new coupling transformation;

- We show that only minor architectural changes are required to switch between between generative and discriminative configurations, while maintaining exact likelihoods and transparent decision mechanisms;

- We demonstrate that MAGIC-Flow outperforms strong generative and discriminative baselines in challenging medical imaging settings, offering a stable, memory-efficient, and interpretable framework for clinically relevant AI.

## 2 THEORETICAL FOUNDATION

Normalizing flows are a class of generative models that represent complex probability distributions by transforming a simple base distribution through a sequence of invertible and differentiable mappings. Let $z \sim p_Z(z)$ be a latent variable and $f$ an invertible mapping. A flow defines $x = f(z)$ and $z = f^{-1}(x)$. The inverse mapping $x \mapsto z$ enables tractable likelihood evaluation, while the forward mapping $z \mapsto x$ enables exact sampling. By the change-of-variables formula, the log-likelihood of $x$ is

$$\log p_X(x) = \log p_Z(f^{-1}(x)) + \log \left| \det \frac{\partial f^{-1}(x)}{\partial x} \right|. \tag{1}$$

The design of flow transformations balances two requirements: (i) *expressiveness*, to capture complex distributions, and (ii) *tractability*, to allow efficient computation of inverses and Jacobian determinants. Several architectures embody these trade-offs: NICE (Dinh et al., 2014) introduced additive coupling layers; RealNVP (Dinh et al., 2016) extended this with affine couplings and multiscale architectures; Glow (Kingma & Dhariwal, 2018) further improved expressiveness with invertible $1 \times 1$ convolutions. See Kobyzev et al. (2020); Papamakarios et al. (2021) for surveys.

Normalizing flows therefore provide a solid framework for generative modeling, combining exact likelihood training with efficient sampling. We now extend this formulation to the *conditional* setting, which is central to our architecture.

**Conditional normalizing flows.** Conditional normalizing flows (cNFs) incorporate auxiliary information $y$ (e.g., class labels or embeddings) into the transformation $f(\cdot, y)$, enabling modeling of conditional densities $p_{X|Y}$. Applications include structured prediction (Winkler et al., 2019), guided image generation with conditional invertible neural networks (cINNs) (Ardizzone et al., 2019), and autoregressive conditional flows (CAFLOW) (Batzolis et al., 2021). More recent works apply cNFs to mitigate mode collapse (Kanaujia et al., 2024) or enable anomaly detection (Gudovskiy et al., 2022).

Our theoretical contribution is to show that the two cornerstone properties of flows—*invertibility* and *Jacobian factorization*—also hold in the conditional setting. Extensive proofs are provided in Appendix A.1 and A.2.

**Invertibility property.** Under mild assumptions, the conditional change-of-variable formula is

$$p_{X|Y}(x \mid y) = p_Z\big(f^{-1}(x, y)\big) \cdot \left| \det \frac{\partial f^{-1}(x, y)}{\partial x} \right|. \tag{2}$$

Here, $x \in \mathbb{R}^{C \times H \times W}$ denotes images, $y \in \mathbb{R}^K$ encodes labels (e.g., one-hot or learned embeddings), and $z \sim \mathcal{N}(0, I)$ is the latent representation. Conditioning is handled by parameterizing each flow transformation $f_i(\cdot, y)$ with $y$, while preserving bijectivity in $x$.

**Factorization property.** As in the unconditional case, the overall flow can be expressed as a composition of $N$ bijective mappings $f = f_N \circ f_{N-1} \circ \ldots \circ f_1$ with intermediate states $h_i = f_i(h_{i-1}, y)$, where $h_0 = x$ and $h_N = z$. Since each $f_i(\cdot, y)$ is bijective in $x$, their composition is bijective as well. The Jacobian determinant factorizes across layers:

$$\left| \det \frac{\partial f^{-1}(x, y)}{\partial x} \right| = \prod_{i=1}^{N} \left| \det \frac{\partial f_i^{-1}(h_i, y)}{\partial h_i} \right|. \tag{3}$$

This result establishes that conditional flows inherit the same tractability guarantees as unconditional flows, while enabling controlled generation and likelihood-based classification. It forms the theoretical foundation of MAGIC-Flow, which builds upon conditional invertibility and factorization to support both generative and discriminative tasks within a shared architecture.

## 3 Unified Architecture of MAGIC-Flow

Building on the conditional formulation from the previous section, we present the unified architecture of MAGIC-Flow. The theoretical foundation showed that invertibility and Jacobian factorization extend naturally to the conditional case, guaranteeing that conditional flows remain tractable. MAGIC-Flow instantiates these principles in a hierarchical, multiscale design that combines conditional flow steps, squeeze operations, and split operations. This framework is fully general: only the definition of the affine coupling transformation is task-specific, as discussed later in Section 4.

**Flow steps.** A single conditional flow step (Figure 1a) consists of three standard invertible components: (1) **ActNorm** (Kingma & Dhariwal, 2018): a channel-wise affine transformation initialized from data statistics (Salimans & Kingma, 2016) and subsequently learned as trainable parameters; (2) **Invertible** $1 \times 1$ **Convolution** (Kingma & Dhariwal, 2018): a learnable channel permutation that improves flexibility while preserving invertibility and efficient Jacobian computation; (3) **Affine Coupling Transformation** (Dinh et al., 2016): the core operation, here extended to incorporate conditioning on $y$. The scale and shift functions of the coupling transformation are defined by task-specific layers (Section 4).

Concretely, the input $\mathbf{x} \in \mathbb{R}^{C \times H \times W}$ is partitioned into $\mathbf{x}_A$ and $\mathbf{x}_B$ with a binary mask $\mathbf{M}$, such that $\mathbf{x}_A = \mathbf{M} \odot \mathbf{x}$ and $\mathbf{x}_B = (1 - \mathbf{M}) \odot \mathbf{x}$. The two partitions are alternately transformed, conditioned on the other partition and the label $y$, via scale and shift functions $\mathbf{s}_i, \mathbf{t}_i$:

$$\mathbf{u}_A = \mathbf{x}_A, \quad \mathbf{u}_B = \mathbf{x}_B \odot \exp(\mathbf{s}_1(\mathbf{x}_A, y)) + \mathbf{t}_1(\mathbf{x}_A, y),$$
$$\mathbf{x}'_A = \mathbf{u}_A \odot \exp(\mathbf{s}_2(\mathbf{u}_B, y)) + \mathbf{t}_2(\mathbf{u}_B, y), \quad \mathbf{x}'_B = \mathbf{u}_B.$$

The output is recombined as $\mathbf{x}' = \mathbf{x}'_A \oplus \mathbf{x}'_B$, with log-determinant

$$\log |\det J| = \sum_{c,h,w} (1 - M_{c,h,w})\, s_{1,c,h,w} + \sum_{c,h,w} M_{c,h,w}\, s_{2,c,h,w}.$$

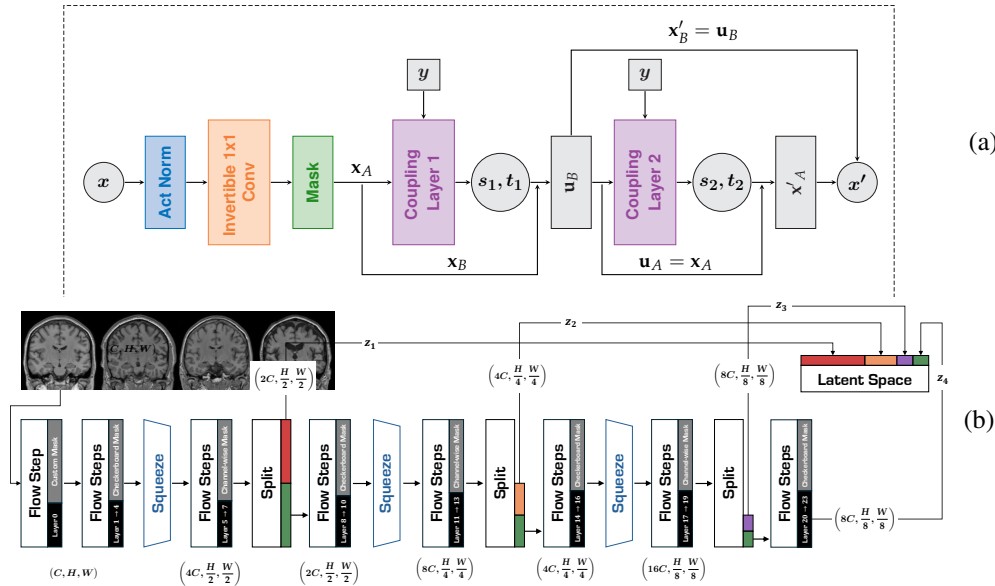

Figure 1: Illustration of components of the model: (a) single flow step and (b) full multiscale flow.

**Squeeze and split operations.** To model multiscale structure efficiently, MAGIC-Flow incorporates two standard architectural operations: *Squeeze* that reshapes a tensor of size $(C, H, W)$ into $(4C, H/2, W/2)$, reducing spatial resolution while increasing channel depth. This allows coupling layers to operate over larger receptive fields without added convolutional cost (Hoogeboom et al., 2019); *Split* that divides the feature map along the channel dimension into two halves. One half is factored out directly into the latent representation, while the other continues through subsequent flow steps. Splits yield a hierarchical latent variable $z$ that encodes fine-grained details at early stages and coarser structure at later stages (Ardizzone et al., 2019; Gudovskiy et al., 2022).

**Masking strategy.** To ensure that all dimensions of $\mathbf{x}$ are transformed across the flow, MAGIC-Flow alternates between three types of binary masks $\mathbf{M} \in \{0,1\}^{C \times H \times W}$: (1) **Checkerboard masks:** update alternating spatial locations to promote pixel-level mixing; (2) **Channel-wise masks:** transform subsets of channels, enabling feature-level transformations; (3) **Application-specific masks:** selectively emphasize semantically relevant regions, adapting the transformation to the downstream objective. Alternating checkerboard and channel-wise masks across flow steps balances spatial and feature-level expressiveness while maintaining tractable Jacobian computations. See Appendix A.3 for details and visualization.

**Overall design.** The full MAGIC-Flow architecture (Figure 1b) stacks 24 conditional flow steps organized in a multiscale hierarchy. The sequence begins with a flow step using an application-specific mask. Each block applies three checkerboard flow steps, a squeeze operation, three channel-wise steps, and then a split, where half of the channels are routed to the latent space while the remainder continue through deeper transformations. Repeating this pattern yields a hierarchical latent decomposition that captures both local and global structure while maintaining tractable likelihoods. This design leverages conditional invertibility, multiscale factorization, and structured masking to support both expressive generation and efficient likelihood-based classification within a single unified framework. The only task-specific component is the affine coupling transformation, detailed in Section 4.

## 4 TASK-SPECIFIC COUPLING LAYERS AND LEARNING OBJECTIVES

While the unified architecture of MAGIC-Flow (Section 3) guarantees conditional invertibility and tractable likelihood computation, its flexibility stems from the design of the *task-specific affine cou-*

*pling layers*. These layers specialize the shared framework for two complementary objectives: high-fidelity conditional image generation and accurate label-informed classification.

## 4.1 TASK-SPECIFIC AFFINE COUPLING LAYERS

Both coupling layers adopt a convolutional conditioning network that maps the input $\mathbf{x} \in \mathbb{R}^{C \times H \times W}$ and label $\mathbf{y} \in \mathbb{R}^K$ into scale and shift parameters $(s, t)$. The key difference lies in how conditioning is injected and how feature transformations are optimized for each task. See Appendix A.4 for detailed architectures of both layers.

**Generation coupling layer.** For conditional synthesis, the coupling network prioritizes *expressiveness* and *contextual richness*. Conditioning information is injected at multiple depths via FiLM layers (Perez et al., 2018), while CBAM modules (Woo et al., 2018) provide channel- and spatial-wise attention. Residual blocks with FiLM modulation (He et al., 2016) enhance representational capacity. To capture long-range dependencies and multi-scale context, we integrate global context blocks (Hu et al., 2018; Cao et al., 2019) and ASPP-SE modules (Chen et al., 2017; Hu et al., 2018). Together, these components enable the coupling transformation to leverage label information across scales, producing condition-aware transformations richer and more flexible than conventional shallow coupling networks (Dinh et al., 2016; Kingma & Dhariwal, 2018).

**Classification coupling layer.** For discriminative modeling, the coupling network is streamlined to emphasize *label-informed, discriminative transformations* rather than multi-scale expressiveness required for generation. It includes a residual label embedding module, FiLM-based conditioning, CBAM attention, and residual convolutional blocks. Features are normalized, regularized with dropout, and fused with the label embedding, and then passed to parallel heads that predict translation and bounded scale parameters. Although the more expressive generative coupling could in principle be reused for classification, our ablations show that the lightweight discriminative variant maintains accuracy while avoiding unnecessary capacity, yielding comparable performance with substantially lower computational and memory cost. This design enables explicitly label-conditioned transformations that are robust for likelihood-based classification.

## 4.2 GENERATION AND CLASSIFICATION PROCEDURES

With the task-specific couplings defined, we now describe how MAGIC-Flow performs the two tasks: image generation and classification. Both tasks leverage the same invertible backbone, differing only in the coupling transformation.

**Training objective.** Given a dataset of paired inputs and labels $\{(x_i, y_i)\}_{i=1}^N$, the parameters of MAGIC-Flow are optimized by maximizing the conditional log-likelihood:

$$\mathcal{L}(\theta) = \frac{1}{N} \sum_{i=1}^N \log p_{X|Y}(x_i \mid y_i) = \frac{1}{N} \sum_{i=1}^N \left[ \log p_Z(f_\theta^{-1}(x_i, y_i)) + \log \left| \det \frac{\partial f_\theta^{-1}(x_i, y_i)}{\partial x_i} \right| \right].$$

**Generation.** To synthesize an image conditioned on a label $\mathbf{y}$, we draw a latent variable $\mathbf{z} \sim \mathcal{N}(0, I)$ and apply the inverse flow: $\mathbf{x} = f^{-1}(\mathbf{z}, \mathbf{y})$. The hierarchical architecture (squeeze, split, and multiscale flow steps) ensures that both global structure and fine detail are generated consistently. The generation coupling layers inject label information at every stage, enabling the synthesis of diverse and realistic images aligned with $\mathbf{y}$.

**Classification.** To classify an input $\mathbf{x}$, MAGIC-Flow evaluates conditional likelihoods. The forward flow maps the input into the latent space $\mathbf{z} = f(\mathbf{x}, \mathbf{y})$ and the conditional log-likelihood can be computed using Equation 2. At inference time, the predicted label $\hat{\mathbf{y}}$ is obtained by maximum likelihood:

$$\hat{\mathbf{y}} = \arg\max_{\mathbf{y}} \log p_{X|Y}(\mathbf{x} \mid \mathbf{y}).$$

## 5 Interpretability via Likelihood Attribution Maps

A key advantage of MAGIC-Flow over existing generative frameworks is its ability to compute *exact likelihoods*. This property enables not only training via maximum likelihood estimation, but also interpretability through likelihood decomposition. We introduce *likelihood attribution maps*, which quantify the spatial contribution of each image location to the conditional log-likelihood. Unlike heuristic, post-hoc methods such as Grad-CAM (Selvaraju et al., 2017) or integrated gradients (Sundararajan et al., 2017), our attribution maps are derived directly from the internal computations of the flow, ensuring faithfulness and model consistency. To compute $\mathcal{H}(\mathbf{x}, \mathbf{y}) \in \mathbb{R}^{C \times H \times W}$, we traverse the model as follows:

1. **Forward pass:** Map a input $\mathbf{x}$ conditioned on $\mathbf{y}$ to the latent representation $\mathbf{z}$ via the conditional flow.

2. **Backward pass:** Accumulate contributions to the log-likelihood from two sources:
   - *Local Jacobian contributions:* Each affine coupling layer $i$ contributes
   $$\ell_J^i(\mathbf{x}, \mathbf{y})_{c,h,w} = (1 - M_{c,h,w}) \, s_{1,c,h,w}(\mathbf{x}_A, \mathbf{y}) + M_{c,h,w} \, s_{2,c,h,w}(\mathbf{u}_B, \mathbf{y}).$$
   - *Latent contributions:* In the multiscale architecture, split operations factor out subsets $\mathbf{z}_j$ ($j = 1, \ldots, S + 1$, where $S$ is the number of splits), each contributing via its Gaussian prior $\log p(\mathbf{z}_j)_{c,h,w}$.

3. **Accumulation:** Starting from the final latent subset ($\mathbf{z}_4$ in Figure 1b, size $(8C, H/8, W/8)$), we add its log-probability $\log p(\mathbf{z}_4)$ to the Jacobian contributions of the last flow layers ($i = 20, \ldots, 23$), which share the same resolution. At the subsequent split, we concatenate this attribution map with the factored-out latent part ($\mathbf{z}_3$, also $(8C, H/8, W/8)$) along the channel dimension, yielding a combined attribution map of size $(16C, H/8, W/8)$. Continuing backward, we add the Jacobian contributions from layers $i = 17, \ldots, 19$, then *unsqueeze* the map to $(4C, H/4, W/4)$ to align with the earlier stage (see Figure 6 in Appendix A.5 for a detailed illustration of these initial steps). This process of summation, concatenation, and reshaping is repeated until the full attribution map $\mathcal{H}(\mathbf{x}, \mathbf{y})$ is obtained.

This recursive construction yields a faithful, spatially resolved map of how each pixel contributes to the conditional log-likelihood, providing insight into MAGIC-Flow's reasoning while fully respecting the model's invertible structure.

## 6 Experiments

We evaluate MAGIC-Flow on two tasks: *conditional generation* (scanner- and modality-conditioned) and *classification* (scanner and disease identification). Each task is tested on diverse, publicly available datasets, compared against strong baselines, and evaluated with task-appropriate metrics. All datasets and preprocessing details are provided in Appendix B.1 and B.2.

### 6.1 Generation Experiments

**Tasks and Datasets.** We assess two conditional generation tasks: (i) scanner-conditioned generation, evaluating whether MAGIC-Flow can synthesize realistic MRI slices that capture scanner-specific characteristics; and (ii) modality-conditioned generation, testing the ability to generate anatomically consistent images across MRI (T1, T2, FLAIR) and PET (FDG, amyloid, tau) modalities. Scanner-conditioned experiments use T1-weighted MRI slices from PPMI (Marek et al., 2011), IXI (Brain-Development, 2019), and SALD (Wei et al., 2017). Modality-conditioned experiments use multimodal MRI and PET from ADNI3 and ADNI4 (Jack Jr et al., 2008). Dataset statistics are summarized in Table 4a and 4b of Appendix B.2.

**Evaluation Metrics and Benchmarks.** We report FID and KID (Heusel et al., 2017; Bińkowski et al., 2018) with domain-adapted feature extractors (FID$_{\text{Rad}}$ and KID$_{\text{Rad}}$ pre-trained on RadImageNet; FID$_{\text{SwAV}}$ on SwAV) for medical realism (Mei et al., 2022; Caron et al., 2020). To capture sample-level fidelity and diversity, we use PRDC metrics (Precision, Recall, Density, Coverage)

(Kynkäänniemi et al., 2019; Naeem et al., 2020), and intra-/inter-class MS-SSIM (Wang et al., 2003; Odena et al., 2017). Details are provided in Appendix B.3.1 and B.3.2. We compare against GANs (SNGAN (Miyato et al., 2018), StyleGAN2-DiffAug-LeCam (Karras et al., 2020; Zhao et al., 2020; Tseng et al., 2021), ADC-GAN (Hou et al., 2022)), diffusion models (DDPM (Dhariwal & Nichol, 2021) and the medical latent diffusion model MedFusion (Müller-Franzes et al., 2023)), latent-variable models (CVAE (Sohn et al., 2015)), and flow-based models (TARFlow (Zhai et al., 2024) and a custom implementation of conditional RealNVP/Glow baseline, as no publicly available conditional Glow models exist that are suitable for our setting). See Appendix B.4.1 for details.

## 6.2 Classification Experiments

**Task and Dataset.** We evaluate MAGIC-Flow on two classification tasks: (i) scanner classification, where the model discriminates between seven scanners - including scanners with comparable noise profiles such as Siemens Prisma vs. Prisma Fit - using unbalanced datasets to assess robustness; and (ii) Alzheimer's disease (AD) vs. cognitively normal (CN) classification, a clinically relevant downstream task performed on Tau PET from ADNI3/4. Scanner-classification experiments use coronal T1-weighted MRI slices from PPMI, SALD, IXI, and ADNI3. The AD vs. CN task uses coronal Tau PET slices from ADNI3/4. All experiments follow a five-fold cross-validation protocol. Dataset statistics are summarized in Table 4a and 4c of Appendix B.2.

**Evaluation Metrics and Benchmarks.** We report Accuracy, Balanced Accuracy, AUC, and macro-averaged Precision, Recall, and F1-score, accounting for both overall and class-balanced performance. Details are provided in Appendix B.3.3. Attribution maps are shown for the AD vs. CN task, where interpretability is clinically meaningful and helps assess how the model attends to disease-relevant brain regions. Unlike scanner identification - where differences are subtle, spatially diffuse, and interpretability serves mainly to explain model behavior - in diagnosis prediction the attribution maps can highlight anatomically coherent regions of interest, potentially supporting downstream clinical or post-hoc analyses. We compare against CNNs pretrained on RadImageNet (Mei et al., 2022) (ResNet-50, DenseNet-121, InceptionV3, InceptionResNetV2) and Vision Transformers (ViT, ViT-ResNet, Swin Transformer) (Dosovitskiy et al., 2020; Jain et al., 2024; Liu et al., 2021). Baselines are evaluated with and without pretraining. See Appendix B.4.2 for details.

## 6.3 Quantitative Evaluation of Attribution Maps

To quantitatively assess the interpretability of MAGIC-Flow, we evaluate whether the regions emphasized by our likelihood-based attribution maps in the diagnosis task correspond to pathologically meaningful areas. Each attribution map is segmented at multiple thresholds to obtain regions of interest (ROIs) capturing the most salient locations. From each ROI, we extract gray-level intensity-distribution features that reflect Tau tracer uptake, and use these as inputs to logistic-regression classifiers distinguishing CN from AD subjects. By analyzing how classification performance varies across thresholds, we can determine whether MAGIC-Flow concentrates discriminative evidence within clinically relevant regions, rather than relying on non-informative image details.

## 7 Results

In this section, we present the results of the experiments described in Section 6. The generation results are reported in Section 7.1, and the classification results are presented in Section 7.2. Quantitative evaluation of the attribution maps can be found in Section 7.3. Computational and scalability analysis are accessible in Appendix C.1.

## 7.1 Image Generation Results

Quantitative results are summarized in Tables 1 and 2. For scanner-conditioned generation, MAGIC-Flow achieves a $FID_{Rad}$ of 0.84 and a $KID_{Rad}$ of $6.7 \times 10^{-4}$, surpassing the closest baseline, Med-Fusion (1.43 and $1.3 \times 10^{-3}$), with non-overlapping 95% confidence intervals confirming statistical significance. In modality-conditioned generation, MAGIC-Flow yields comparable improvements ($FID_{Rad}$ 0.98, $KID_{Rad}$ $4.5 \times 10^{-4}$), with bootstrapped intervals indicating increased stable performance.

Table 1: Comparison of generative models on scanner- and modality-conditioned tasks using FID variants. Bold indicates best scores; 95% CIs are shown for $FID_{Rad}$ and $KID_{Rad}$.

| Task | Model | FID $\downarrow$ | $FID_{Rad}\downarrow$ | $KID_{Rad}\downarrow$ | $FID_{SwAV}\downarrow$ | $FID^b_{Rad}$ (CI$_{95\%}$) $\downarrow$ | $KID^b_{Rad}$ (CI$_{95\%}$) $\downarrow$ |
|---|---|---|---|---|---|---|---|
| **Scanner** (*Generation*) | MAGIC-Flow | 27.64 | **0.84** | **6.7** $\times\,10^{-4}$ | 4.31 | **[0.83, 0.95]** | **[6.2, 7.5]** $\times\,10^{-4}$ |
| | SNGAN | 29.91 | 3.03 | 2.4 $\times\,10^{-3}$ | 4.98 | [2.95, 3.18] | [2.4, 2.7] $\times\,10^{-3}$ |
| | StyleGAN2 | 97.50 | 6.80 | 6.7 $\times\,10^{-3}$ | 9.33 | [6.64, 6.95] | [6.5, 6.9] $\times\,10^{-3}$ |
| | ADC-GAN | 59.83 | 5.12 | 4.6 $\times\,10^{-3}$ | 5.21 | [5.05, 5.31] | [4.5, 4.8] $\times\,10^{-3}$ |
| | DDPM | 39.5 | 13.80 | 1.75 $\times\,10^{-2}$ | 5.74 | [13.63, 14.01] | [1.73, 1.78] $\times\,10^{-2}$ |
| | MedFusion | **18.73** | 1.43 | 1.3 $\times\,10^{-3}$ | **2.94** | [1.41, 1.51] | [1.23, 1.39] $\times\,10^{-3}$ |
| | CVAE | 244.07 | 16.69 | 2.07 $\times\,10^{-2}$ | 21.56 | [16.58, 16.83] | [2.06, 2.10] $\times\,10^{-2}$ |
| | Glow | 151.09 | 4.03 | 4.42 $\times\,10^{-3}$ | 14.49 | [3.96, 4.16] | [4.3, 4.5] $\times\,10^{-4}$ |
| | TarFlow | 71.31 | 3.68 | 4.14 $\times\,10^{-3}$ | 9.04 | [3.58, 3.83] | [3.9, 4.4] $\times\,10^{-4}$ |
| **Modality** (*Generation*) | MAGIC-Flow | 33.04 | **0.98** | **4.5** $\times\,10^{-4}$ | 4.07 | **[0.94, 1.08]** | **[4.0, 5.2]** $\times\,10^{-4}$ |
| | SNGAN | 34.06 | 3.82 | 2.6 $\times\,10^{-3}$ | 4.23 | [3.76, 3.95] | [2.5, 2.7] $\times\,10^{-3}$ |
| | StyleGAN2 | 59.2 | 4.75 | 3.4 $\times\,10^{-3}$ | 6.50 | [4.68, 4.85] | [3.3, 3.5] $\times\,10^{-3}$ |
| | ADC-GAN | 57.84 | 4.94 | 2.7 $\times\,10^{-3}$ | 5.46 | [4.88, 5.07] | [2.4, 2.7] $\times\,10^{-3}$ |
| | DDPM | 70.48 | 15.22 | 1.72 $\times\,10^{-2}$ | 8.08 | [15.04, 15.38] | [1.70, 1.74] $\times\,10^{-2}$ |
| | MedFusion | **23.10** | 1.77 | 9.1 $\times\,10^{-4}$ | **3.54** | [1.72, 1.92] | [8.06, 9.99] $\times\,10^{-4}$ |
| | CVAE | 215.88 | 14.71 | 1.47 $\times\,10^{-2}$ | 19.40 | [14.56, 14.88] | [1.45, 1.48] $\times\,10^{-2}$ |
| | Glow | 155.44 | 11.17 | 1.32 $\times\,10^{-2}$ | 14.55 | [10.97, 11.37] | [1.31, 1.36] $\times\,10^{-2}$ |
| | TarFlow | 74.65 | 11.84 | 1.06 $\times\,10^{-2}$ | 8.93 | [11.69, 12.12] | [1.04, 1.08] $\times\,10^{-2}$ |

Table 2: Comparison of generative models using fidelity/diversity (P, R, D, C) and MS-SSIM. Gray highlights show the best-performing model; "Real Data" provides reference values.

| Task | Model | P $\uparrow$ | R $\uparrow$ | D $\uparrow$ | C $\uparrow$ | MS-SSIM$^{intra}\downarrow$ | MS-SSIM$^{inter}\downarrow$ |
|---|---|---|---|---|---|---|---|
| **Scanner** (*Generation*) | Real Data | - | - | - | - | $0.51 \pm 0.11$ | $0.43 \pm 0.11$ |
| | MAGIC-Flow | 0.87 | 0.64 | 0.91 | 0.84 | $0.60 \pm 0.08$ | $0.49 \pm 0.09$ |
| | SNGAN | 0.79 | 0.05 | 0.77 | 0.40 | $0.68 \pm 0.16$ | $0.48 \pm 0.09$ |
| | StyleGAN2 | 0.99 | 0.0 | 0.61 | 0.04 | $0.99 \pm 0.003$ | $0.66 \pm 0.10$ |
| | ADC-GAN | 0.94 | 0.01 | 0.72 | 0.10 | $0.89 \pm 0.08$ | $0.66 \pm 0.08$ |
| | DDPM | 0.0 | 1.0 | 0.0 | 0.0 | $0.46 \pm 0.12$ | $0.33 \pm 0.12$ |
| | MedFusion | 0.97 | 0.47 | 0.75 | 0.64 | $0.56 \pm 0.18$ | $0.38 \pm 0.16$ |
| | CVAE | 0.0 | 0.0 | 0.0 | 0.0 | $0.99 \pm 0.002$ | $0.83 \pm 0.06$ |
| | Glow | 0.02 | 0.13 | 0.0 | 0.01 | $0.34 \pm 0.09$ | $0.30 \pm 0.07$ |
| | TarFlow | 0.26 | 0.30 | 0.11 | 0.20 | $0.55 \pm 0.09$ | $0.50 \pm 0.08$ |
| **Modality** (*Generation*) | Real Data | - | - | - | - | $0.61 \pm 0.12$ | $0.22 \pm 0.18$ |
| | MAGIC-Flow | 0.94 | 0.64 | 1.0 | 0.87 | $0.64 \pm 0.09$ | $0.22 \pm 0.18$ |
| | SNGAN | 0.89 | 0.0 | 0.98 | 0.33 | $0.79 \pm 0.18$ | $0.22 \pm 0.19$ |
| | StyleGAN2 | 0.98 | 0.02 | 1.0 | 0.25 | $0.90 \pm 0.07$ | $0.29 \pm 0.25$ |
| | ADC-GAN | 1.0 | 0.0 | 0.87 | 0.17 | $0.87 \pm 0.11$ | $0.23 \pm 0.21$ |
| | DDPM | 0.0 | 0.0 | 0.0 | 0.0 | $0.55 \pm 0.12$ | $0.20 \pm 0.18$ |
| | MedFusion | 0.98 | 0.48 | 1.0 | 0.74 | $0.63 \pm 0.15$ | $0.19 \pm 0.18$ |
| | CVAE | 0.0 | 0.0 | 0.0 | 0.0 | $0.99 \pm 0.003$ | $0.32 \pm 0.29$ |
| | Glow | 0.26 | 0.05 | 0.05 | 0.01 | $0.26 \pm 0.07$ | $0.20 \pm 0.06$ |
| | TarFlow | 0.10 | 0.31 | 0.03 | 0.03 | $0.33 \pm 0.17$ | $0.30 \pm 0.18$ |

GAN-based baselines exhibit the typical fidelity–diversity trade-off (e.g., SNGAN: P=0.79, R=0.05, D=0.77, C=0.40), whereas flow-based models generally underperform in both realism and sample variety (e.g., TarFlow: P=0.26, R=0.30, D=0.11, C=0.20). Among diffusion-based methods, DDPM shows limited performance across all metrics. MedFusion achieves slightly higher precision and

comparable - or slightly lower - density than MAGIC-Flow, but it remains less diverse, as reflected in its lower recall and coverage. Metrics computed with feature extractors trained on natural images (FID and $FID_{SwAV}$) favor MedFusion, although these backbones are less aligned with the imaging domain and can yield less reliable evaluations. MAGIC-Flow, in contrast, achieves the most balanced fidelity-diversity profile, with high precision, recall, density, and coverage (scanner: 0.87 / 0.64 / 0.91 / 0.84; modality: 0.94 / 0.64 / 1.0 / 0.87). MS-SSIM further confirms that MAGIC-Flow preserves realistic intra-class variability while maintaining inter-class separability close to that of real data. Qualitative comparisons can be found in Figure 7 of Appendix C.2.

Finally, MAGIC-Flow demonstrates a strong quality–throughput trade-off, generating 37.5 images/s, substantially faster than the strongest baseline in terms of output quality, MedFusion (1.77 images/s), and competitive with other generative models (Glow: 300.0; TarFlow: 0.11; SNGAN: 49.75; ADC-GAN: 38.17; StyleGAN2: 52.20; CVAE: 369.0; DDPM: 0.24). Detailed efficiency analysis—including training steps, compute time, memory usage, and per-step cost—is provided in Appendix C.1.

## 7.2 IMAGE CLASSIFICATION RESULTS

For scanner classification, as shown in Table 3, MAGIC-Flow achieves $0.90 \pm 0.01$ accuracy, comparable to CNN and ViT baselines, while surpassing them in balanced accuracy ($0.76 \pm 0.02$), macro recall ($0.76 \pm 0.03$), and F1-score ($0.75 \pm 0.03$), demonstrating enhanced robustness to underrepresented scanners and discrimination of *close* classes. For diagnosis classification, MAGIC-Flow also reaches $0.90 \pm 0.01$ accuracy, similarly to the strongest baselines, while improving balanced accuracy ($0.81 \pm 0.06$) and maintaining competitive macro recall ($0.81 \pm 0.06$) and F1-score ($0.80 \pm 0.02$).

The likelihood-based attribution maps for the AD vs. CN task produced by MAGIC-Flow (Figure 2) identify regions most informative for the model's decisions. High-likelihood (red) areas correspond to commonly observed features, while low-likelihood (blue) regions highlight atypical patterns that deviate from the overall population. Notably, in the context of the AD vs. CN task, these low-likelihood regions correspond precisely to tau aggregates, providing an interpretable explanation of the pathological features driving the model's predictions.

Table 3: Comparison of classification performance on the test set for both generation tasks.

| Task | Model | PreTrained | Accuracy | Bal. Acc. | AUC | Precision | Recall | F1-score |
|---|---|---|---|---|---|---|---|---|
| **Scanner** *(Classification)* | MAGIC-Flow | - | $0.90 \pm 0.01$ | $0.76 \pm 0.02$ | $0.97 \pm 0.01$ | $0.77 \pm 0.03$ | $0.76 \pm 0.03$ | $0.75 \pm 0.03$ |
| | ResNet-50 | RadImageNet | $0.91 \pm 0.01$ | $0.73 \pm 0.02$ | $0.98 \pm 0.00$ | $0.71 \pm 0.06$ | $0.73 \pm 0.02$ | $0.71 \pm 0.04$ |
| | DenseNet-121 | RadImageNet | $0.91 \pm 0.01$ | $0.73 \pm 0.03$ | $0.98 \pm 0.00$ | $0.79 \pm 0.03$ | $0.73 \pm 0.03$ | $0.72 \pm 0.03$ |
| | InceptionV3 | RadImageNet | $0.90 \pm 0.02$ | $0.72 \pm 0.02$ | $0.97 \pm 0.00$ | $0.67 \pm 0.04$ | $0.72 \pm 0.02$ | $0.69 \pm 0.03$ |
| | IncepResNetV2 | RadImageNet | $0.88 \pm 0.02$ | $0.74 \pm 0.03$ | $0.97 \pm 0.01$ | $0.76 \pm 0.03$ | $0.74 \pm 0.03$ | $0.74 \pm 0.03$ |
| | ViT | - | $0.88 \pm 0.01$ | $0.70 \pm 0.02$ | $0.97 \pm 0.01$ | $0.72 \pm 0.02$ | $0.70 \pm 0.02$ | $0.70 \pm 0.02$ |
| | ViT-ResNet | ImageNet21k | $0.91 \pm 0.01$ | $0.73 \pm 0.03$ | $0.98 \pm 0.00$ | $0.74 \pm 0.09$ | $0.73 \pm 0.03$ | $0.71 \pm 0.04$ |
| | Swin-ViT | ImageNet22k | $0.91 \pm 0.00$ | $0.73 \pm 0.03$ | $0.98 \pm 0.01$ | $0.71 \pm 0.07$ | $0.73 \pm 0.03$ | $0.71 \pm 0.05$ |
| **Diagnosis** *(Classification)* | MAGIC-Flow | - | $0.90 \pm 0.01$ | $0.81 \pm 0.05$ | $0.89 \pm 0.04$ | $0.81 \pm 0.02$ | $0.81 \pm 0.05$ | $0.80 \pm 0.02$ |
| | ResNet-50 | RadImageNet | $0.88 \pm 0.03$ | $0.79 \pm 0.07$ | $0.87 \pm 0.04$ | $0.76 \pm 0.04$ | $0.79 \pm 0.07$ | $0.77 \pm 0.05$ |
| | DenseNet-121 | RadImageNet | $0.89 \pm 0.02$ | $0.80 \pm 0.05$ | $0.91 \pm 0.01$ | $0.79 \pm 0.03$ | $0.80 \pm 0.05$ | $0.79 \pm 0.03$ |
| | InceptionV3 | RadImageNet | $0.87 \pm 0.03$ | $0.76 \pm 0.06$ | $0.88 \pm 0.05$ | $0.74 \pm 0.04$ | $0.76 \pm 0.06$ | $0.75 \pm 0.04$ |
| | IncepResNetV2 | RadImageNet | $0.88 \pm 0.01$ | $0.72 \pm 0.07$ | $0.85 \pm 0.05$ | $0.76 \pm 0.03$ | $0.72 \pm 0.07$ | $0.73 \pm 0.05$ |
| | ViT | - | $0.90 \pm 0.02$ | $0.79 \pm 0.04$ | $0.92 \pm 0.05$ | $0.82 \pm 0.05$ | $0.79 \pm 0.04$ | $0.80 \pm 0.03$ |
| | ViT-ResNet | ImageNet21k | $0.90 \pm 0.02$ | $0.78 \pm 0.10$ | $0.90 \pm 0.06$ | $0.85 \pm 0.07$ | $0.78 \pm 0.10$ | $0.78 \pm 0.07$ |
| | Swin-ViT | ImageNet22k | $0.90 \pm 0.02$ | $0.81 \pm 0.05$ | $0.91 \pm 0.04$ | $0.82 \pm 0.07$ | $0.81 \pm 0.05$ | $0.80 \pm 0.03$ |

## 7.3 QUANTITATIVE EVALUATION OF ATTRIBUTION MAPS

Figure 3 shows the relationship between attribution-map thresholding and classification performance (F1 score) for the AD vs. CN task. Performance peaks at a threshold of 0.8 (F1 = 0.88; ~40% area

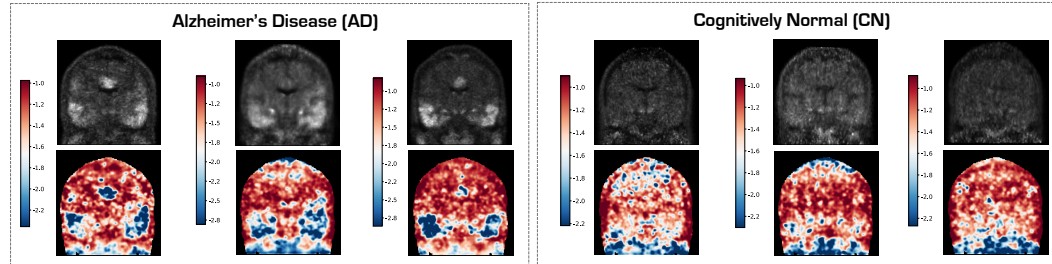

Figure 2: Likelihood attribution maps from MAGIC-Flow. Blue, low-likelihood regions highlight atypical or anomalous inputs, offering explainability, while red, high-likelihood areas correspond to frequently observed features.

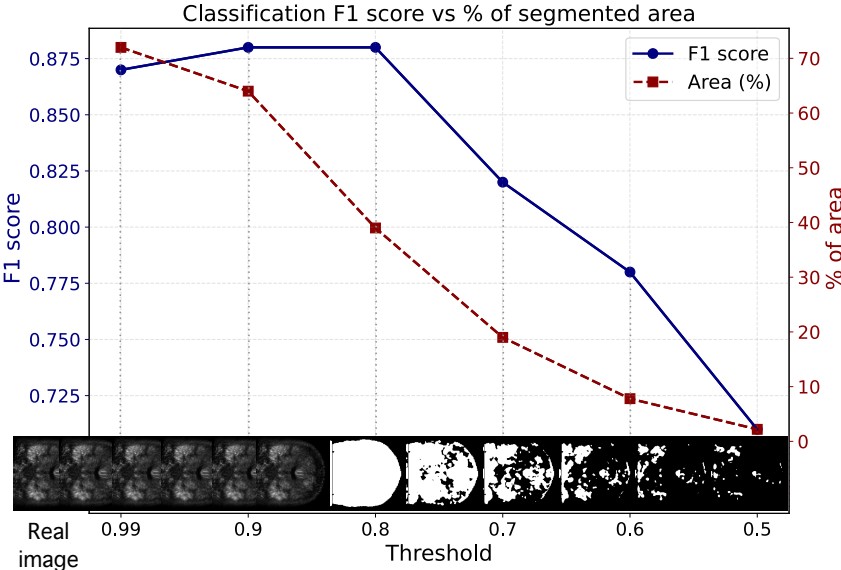

Figure 3: Attribution map evaluation via threshold–performance analysis. The F1 score (blue) and retained area (red) are shown as functions of the attribution-map threshold, with the corresponding Tau-PET segmentations for an example image displayed below (bottom).

retained), revealing a non-monotonic trend: overly inclusive thresholds (e.g., 0.99, retaining 72% of the area) dilute the discriminative signal (F1 = 0.87), whereas overly restrictive thresholds (e.g., 0.5, retaining ∼2% area) discard informative regions (F1 = 0.71). As illustrated in the bottom panels, intermediate thresholds isolate spatially coherent regions corresponding to cortical areas known to accumulate Tau pathology in AD, while excessively high thresholds fragment the signal into disconnected clusters. These findings provide quantitative evidence that MAGIC-Flow allocates probability mass to pathologically meaningful regions, supporting the interpretability and clinical relevance of our approach.

## 8 CONCLUSION

We introduced MAGIC-Flow, a conditional multiscale normalizing flow that unifies generation and classification within a likelihood-based framework. It enables high-fidelity synthesis, robust classification under scanner variability and imbalance, and interpretable likelihood attribution maps. Current limitations include its 2D scope; extending to 3D, adding uncertainty measures, and exploring pathology-conditioned generation and multi-institutional deployment are future directions. A key use is augmenting scarce datasets of underrepresented conditions. MAGIC-Flow advances joint generative-discriminative modeling for data-limited, privacy-sensitive domains.

REPRODUCIBILITY STATEMENT

Extensive theoretical proof are provided in Appendix A.1 and A.2 while architectural details in A.4, A.3 and A.5. We provide a comprehensive description of datasets and preprocessing in Appendix B.1 and B.2. The evaluation procedures are provided in B.3. A complete overview of baseline models and their implementations are provided Appendix B.4. All code and trained models will be released anonymously as supplementary material. We believe that these resources ensure that all results in this paper are reproducible.

USE OF LLM

This paper has benefited from the use of a Large Language Model (LLM), which was employed solely to aid or polish the writing. The research ideas, methodology, and results are entirely the authors' own.

CODE OF ETHICS

This study involves only publicly available datasets containing human subjects. No private, sensitive, or personally identifiable information has been collected or processed by the authors. All datasets are properly cited in the paper, and the links and footnotes referring to them have been carefully verified to ensure correctness.

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

# SUPPLEMENTARY MATERIAL

## A  MATHEMATICAL AND ARCHITECTURAL DETAILS

### A.1  CONDITIONAL CHANGE OF VARIABLES FORMULA: DETAILED DERIVATION

Let $Z \in \mathbb{R}^d$ and $Y \in \mathbb{R}^k$ be random variables with joint density $p_{Z,Y}(z, y)$. Let

$$f : \mathbb{R}^d \times \mathbb{R}^k \to \mathbb{R}^d$$

be continuously differentiable, and suppose that for every fixed $y$, the map $z \mapsto f(z, y)$ is a diffeomorphism onto its image with Jacobian matrix $\frac{\partial f}{\partial z}(z, y)$ having nonzero determinant. Define $X := f(Z, Y)$, and write $f^{-1}(x, y)$ for the unique $z$ such that $f(z, y) = x$. Assume $p_Y(y) > 0$.

If $Z$ and $Y$ are independent, the conditional density of $X$ given $Y = y$ is

$$p_{X|Y}(x \mid y) = p_Z\big(f^{-1}(x, y)\big) \left| \det\!\left(\frac{\partial f^{-1}}{\partial x}(x, y)\right) \right|.$$

**Derivation.**  Consider the mapping

$$(Z, Y) \mapsto (X, Y) = (f(Z, Y), Y),$$

with Jacobian

$$\frac{\partial(x, y)}{\partial(z, y)} = \begin{bmatrix} \frac{\partial f(z,y)}{\partial z} & \frac{\partial f(z,y)}{\partial y} \\ 0 & I_k \end{bmatrix},$$

where $I_k$ is the $k \times k$ identity matrix. Since this is block lower-triangular, its determinant is

$$\left| \det\left(\frac{\partial(x, y)}{\partial(z, y)}\right) \right| = \left| \det\left(\frac{\partial f(z, y)}{\partial z}\right) \right|.$$

By the change-of-variables formula, the joint density of $(X, Y)$ is

$$p_{X,Y}(x, y) = p_{Z,Y}(f^{-1}(x, y), y) \cdot \left| \det\left(\frac{\partial f}{\partial z}(f^{-1}(x, y), y)\right) \right|^{-1}.$$

If $Z$ and $Y$ are independent, then

$$p_{Z,Y}(z, y) = p_Z(z) \cdot p_Y(y).$$

Substituting, we obtain

$$p_{X,Y}(x, y) = p_Z(f^{-1}(x, y)) \cdot p_Y(y) \cdot \left| \det\left(\frac{\partial f}{\partial z}(f^{-1}(x, y), y)\right) \right|^{-1}.$$

The conditional density follows from

$$p_{X|Y}(x \mid y) = \frac{p_{X,Y}(x, y)}{p_Y(y)}.$$

Using the Jacobian determinant of the inverse function,

$$\left| \det\left(\frac{\partial f}{\partial z}(f^{-1}(x, y), y)\right) \right|^{-1} = \left| \det\left(\frac{\partial f^{-1}}{\partial x}(x, y)\right) \right|,$$

we obtain the final expression:

$$p_{X|Y}(x \mid y) = p_Z(f^{-1}(x, y)) \cdot \left| \det\left(\frac{\partial f^{-1}}{\partial x}(x, y)\right) \right|.$$

This result provides the conditional version of the change-of-variables formula used in our model.

## A.2 AFFINE COUPLING: JACOBIAN DERIVATION

Let $\mathbf{x} \in \mathbb{R}^{C \times H \times W}$ be the input image and let $\mathbf{M} \in \{0,1\}^{C \times H \times W}$ be a binary mask. Define the complementary parts

$$\mathbf{x}_A = \mathbf{M} \odot \mathbf{x}, \qquad \mathbf{x}_B = (\mathbf{1} - \mathbf{M}) \odot \mathbf{x}.$$

**First affine transformation ($\mathbf{A} \to \mathbf{B}$).** The update equations are

$$\mathbf{u}_A = \mathbf{x}_A, \qquad \mathbf{u}_B = \mathbf{x}_B \odot \exp\big(\mathbf{s}_1(\mathbf{x}_A; \mathbf{y})\big) + \mathbf{t}_1(\mathbf{x}_A; \mathbf{y}),$$

where $\mathbf{s}_1, \mathbf{t}_1 \in \mathbb{R}^{C \times H \times W}$ are elementwise scale and translation factors.

The Jacobian is

$$J_1 = \frac{\partial(\mathbf{u}_A, \mathbf{u}_B)}{\partial(\mathbf{x}_A, \mathbf{x}_B)}.$$

Since $\mathbf{u}_A = \mathbf{x}_A$, we have

$$\frac{\partial \mathbf{u}_A}{\partial \mathbf{x}_A} = I, \qquad \frac{\partial \mathbf{u}_A}{\partial \mathbf{x}_B} = 0.$$

Each element of $\mathbf{u}_B$ depends elementwise on $x_{B,i}$ via

$$u_{B,i} = x_{B,i} \cdot \exp(s_{1,i}(\mathbf{x}_A; \mathbf{y})) + t_{1,i}(\mathbf{x}_A; \mathbf{y}),$$

so

$$\frac{\partial u_{B,i}}{\partial x_{B,i}} = \exp(s_{1,i}(\mathbf{x}_A; \mathbf{y})).$$

The Jacobian matrix is thus block-triangular:

$$J_1 = \begin{bmatrix} \frac{\partial \mathbf{u}_A}{\partial \mathbf{x}_A} & \frac{\partial \mathbf{u}_A}{\partial \mathbf{x}_B} \\ \frac{\partial \mathbf{u}_B}{\partial \mathbf{x}_A} & \frac{\partial \mathbf{u}_B}{\partial \mathbf{x}_B} \end{bmatrix} = \begin{bmatrix} I & 0 \\ * & \mathrm{diag}\big(\exp(\mathbf{s}_1)\big) \end{bmatrix},$$

where $*$ represents entries that do not affect the determinant. Therefore, the log-determinant of the first transformation is

$$\log|\det J_1| = \sum_{c,h,w} (1 - M_{c,h,w})\, s_{1,c,h,w}.$$

**Second affine transformation ($\mathbf{B} \to \mathbf{A}$).** The second transformation updates $\mathbf{u}_A$ while keeping $\mathbf{u}_B$ fixed:

$$\mathbf{x}'_B = \mathbf{u}_B, \qquad \mathbf{x}'_A = \mathbf{u}_A \odot \exp\big(\mathbf{s}_2(\mathbf{u}_B; \mathbf{y})\big) + \mathbf{t}_2(\mathbf{u}_B; \mathbf{y}).$$

Again, the Jacobian is block-triangular, with

$$\frac{\partial \mathbf{x}'_A}{\partial \mathbf{u}_A} = \mathrm{diag}\big(\exp(\mathbf{s}_2)\big), \qquad \frac{\partial \mathbf{x}'_B}{\partial \mathbf{u}_B} = I.$$

Thus the log-determinant of the second transformation is

$$\log|\det J_2| = \sum_{c,h,w} M_{c,h,w}\, s_{2,c,h,w}.$$

**Total log-determinant.** The two transformations are composed sequentially, so the total log-determinant is

$$\log|\det J| = \log|\det J_1| + \log|\det J_2| = \sum_{c,h,w} (1 - M_{c,h,w})\, s_{1,c,h,w} + \sum_{c,h,w} M_{c,h,w}\, s_{2,c,h,w}.$$

### A.3 MASK DESIGN

To ensure that every dimension of $\mathbf{x}$ is transformed throughout the flow, MAGIC-Flow alternates among three types of binary masks $\mathbf{M} \in \{0,1\}^{C \times H \times W}$: (1) **Checkerboard masks:** update alternating spatial positions to encourage pixel-level mixing; (2) **Channel-wise masks:** transform subsets of channels, enabling feature-level diversity; (3) **Application-specific masks:** emphasize semantically relevant regions, adapting transformations to the downstream task.

By alternating checkerboard and channel-wise masks across flow steps, MAGIC-Flow achieves a balance between spatial and feature-level expressiveness while preserving tractable Jacobian computations.

The masks used in our experiments are shown in Figure 4. Because our applications involve coronal slices, the application-specific mask is designed as a coronal binary mask. We find that incorporating this application-specific mask reduces artifacts in the generated images.

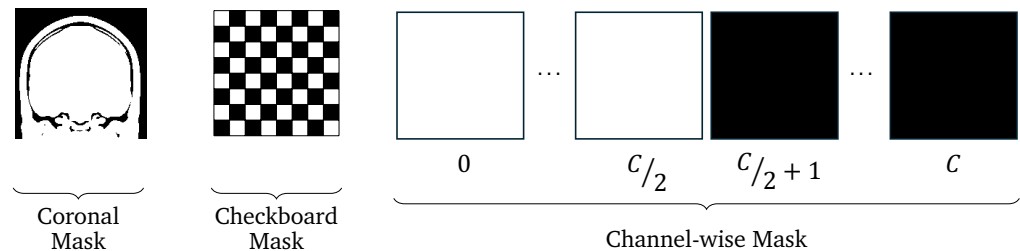

Figure 4: Examples of the masks used in MAGIC-Flow: from left to right, (1) application-specific coronal mask, (2) checkerboard mask, and (3) channel-wise mask. Together, these masks provide complementary spatial, feature-level, and application-adaptive transformations.

### A.4 DESIGN OF AFFINE COUPLING LAYERS

Figure 5 illustrates the task-specific affine coupling layers used in MAGIC-Flow. The generation-oriented coupling layer (Figure 5a) is designed to capture multi-scale dependencies and global context, enabling expressive transformations for high-fidelity sample synthesis. In contrast, the classification-oriented coupling layer (Figure 5b) incorporates label information directly into the feature transformations, prioritizing discriminative structure to improve predictive accuracy. Together, these designs highlight the flexibility of our framework in tailoring flow-based transformations to the demands of different tasks.

### A.5 LIKELIHOOD ATTRIBUTION MAPS

To better understand the predictive behavior of MAGIC-Flow, we introduce *Likelihood Attribution Maps*. These maps provide a pixel-level attribution that links the model's latent likelihood computations to the input domain. As illustrated in Figure 6, the method propagates contributions from deep latent factors back to the input space through inverse split and squeeze operations, while accounting for the Jacobian terms at each stage. The resulting attribution map $\mathcal{H}(\mathbf{x}, \mathbf{y})$ offers an interpretable, likelihood-grounded explanation of the model's output.

## B SUPPLEMENTARY EXPERIMENTAL DETAILS

### B.1 IMAGING PREPROCESSING PIPELINE

All imaging data, including structural MRI and PET scans, were preprocessed using a customized pipeline based on the FMRIB Software Library (FSL) Smith et al. (2004); Jenkinson et al. (2012). T1-weighted images were processed with FSL's fsl_anat pipeline, which included reorientation to standard space, bias-field correction and nonlinear registration to the MNI152 template. T2-weighted and FLAIR images were rigidly coregistered to the subject's T1-weighted image using

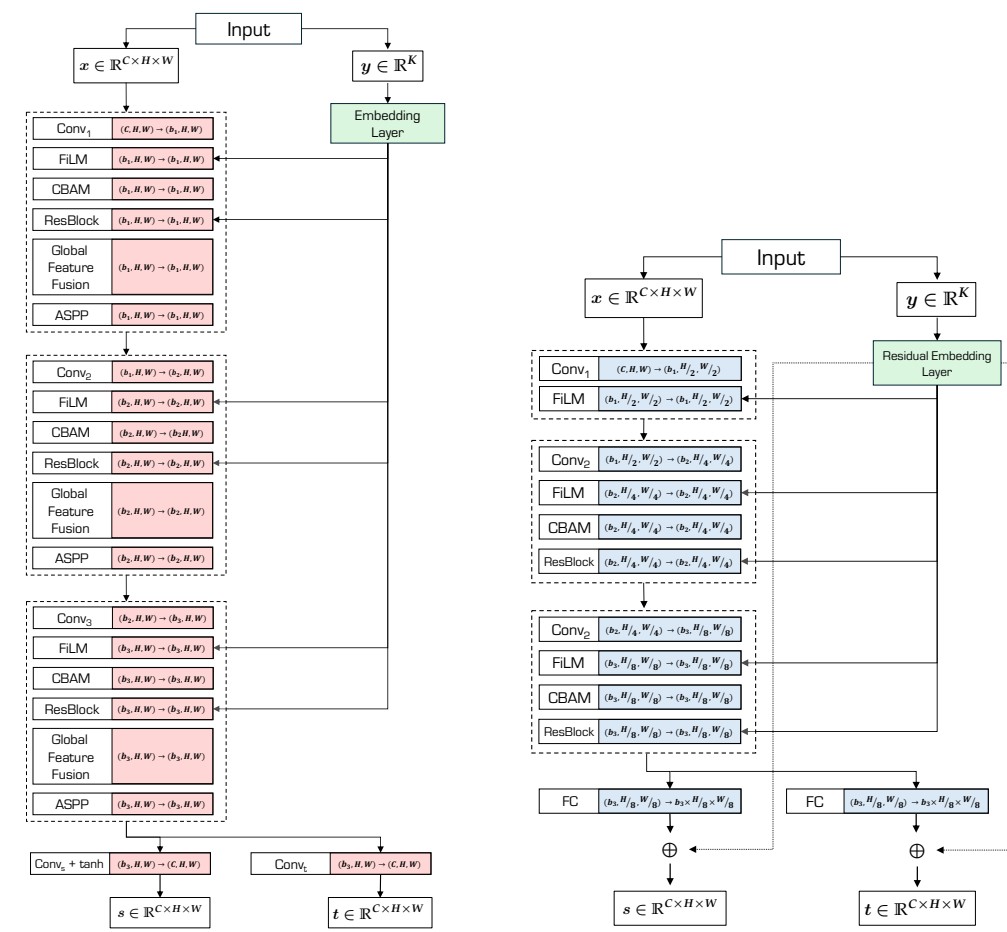

(a) Generation coupling layer: designed for expressive, multi-scale, and context-aware transformations to synthesize high-fidelity outputs.

(b) Classification coupling layer: optimized for label-aware, discriminative feature transformations for predictive accuracy.

Figure 5: Architectures of the task-specific affine coupling layers. The generation layer emphasizes multi-scale and global context features, while the classification layer focuses on integrating label information into the feature transformations.

normalized mutual information cost functions, with or without bias-field correction depending on data quality. The resulting transformations were applied to align T2-weighted and FLAIR scans to T1 space and subsequently to MNI space using the T1-derived warp fields. All PET data were obtained from Alzheimer's Disease Neuroimaging Initiative (ADNI3 and ADNI4). FDG, amyloid, and tau PET scans were first coregistered to the subject's T1-weighted image and then normalized to MNI space using the T1-derived transformations. In ADNI, PET acquisitions occur during the plateau phase of tracer uptake (e.g., 30–60 min post-injection for FDG, 50–70 min for florbetapir and NAV-4694, 90–110 min for florbetaben and MK-6240, 75–105 min for flortaucipir, and 45–75 min for PI-2620) and are reconstructed into multiple short frames. Specifically, florbetapir, florbetaben, and NAV-4694 scans are acquired as $4 \times 5$ min frames, whereas FDG, flortaucipir, and PI-2620 scans are acquired as $6 \times 5$ min frames. To generate a single static image representing tracer distribution, the median across frames was retained. The median was selected over the mean because it provides a more robust estimate of plateau-phase uptake, minimizing the influence of motion artifacts or frame-specific outliers that could bias the averaged signal. This multimodal preprocessing pipeline ensured consistent alignment of structural (T1, T2, FLAIR) and molecular (FDG, amyloid, tau PET) imaging data across participants.

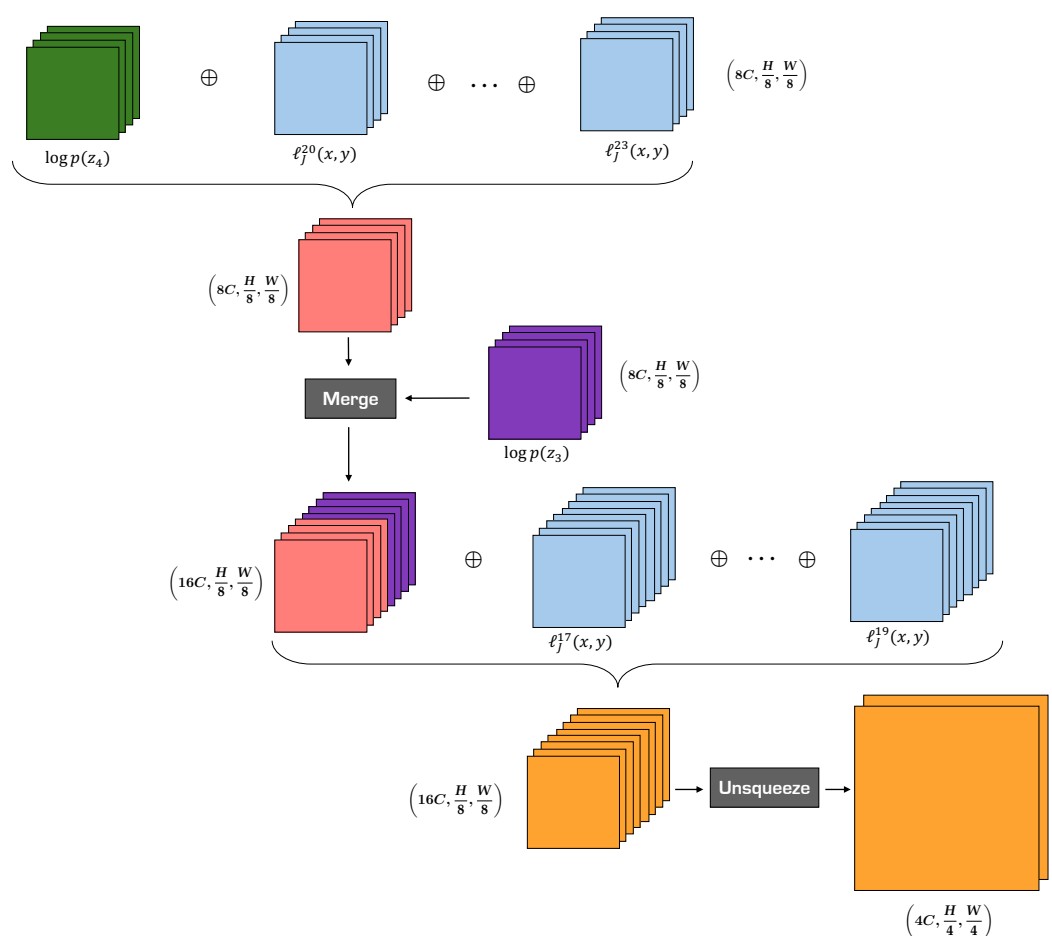

Figure 6: Illustration of the initial steps in constructing the maps described in Section 5. Starting from latent factors (green) at the deepest level, contributions are propagated backward through inverse split and squeeze operations, with the corresponding Jacobian terms accumulated at each step. This process produces a pixel-level attribution map $\mathcal{H}(\mathbf{x}, \mathbf{y})$, providing a principled, likelihood-based interpretation of MAGIC-Flow's predictions.

After preprocessing, the resulting images had dimensions of $(1, 182, 218, 182)$. For our experiments, each 3D volume was processed by extracting coronal slices of shape $(1, 182, 182)$. Specifically, we selected the central coronal slice from each volume. To augment the datasets while maintaining anatomical consistency, neighboring slices were also included only for the scanner-conditioned and modality-conditioned generation tasks, typically within ±5 slices of the center. In some cases, fewer slices were used depending on the dataset size. No slice augmentation was applied for the scanner classification task.

## B.2 IMAGING DATASETS OVERVIEW

### B.2.1 DATASETS FOR GENERATION

Two conditional generation tasks were evaluated: (i) *scanner-conditioned generation*, assessing whether MAGIC-Flow can synthesize realistic MRI slices that preserve scanner-specific characteristics, and (ii) *modality-conditioned generation*, testing the model's ability to generate anatomically consistent images across MRI (T1, T2, FLAIR) and PET (FDG, amyloid, tau) modalities.

Scanner-conditioned experiments used T1-weighted MRI scans from PPMI (Marek et al., 2011), IXI (Brain-Development, 2019), and SALD (Wei et al., 2017), with the following slice counts: PPMI

Table 4: Overview of imaging datasets used across the three tasks. (a) Scanner-conditioned generation and scanner classification: scanner models and manufacturers, along with the number of coronal slices available for each scanner in both classification and generation tasks. (b) Modality-conditioned generation: MRI sequences and PET tracers, with the corresponding number of coronal slices per modality. (a) Diagnosis classification: number of Tau PET coronal slices per diagnostic group.

| Scanner Model | Manufacturer | Dataset | # Coronal slices (Classification) | # Coronal slices (Generation) |
|---|---|---|---|---|
| Gyroscan Intera | Philips | IXI | 322 | 349 |
| Intera | Philips | IXI | 185 | – |
| Unspecified (IOP) | GE | IXI | 74 | 310 |
| TrioTim | SIEMENS | PPMI | 41 | 315 |
| TrioTim | SIEMENS | SALD | 494 | 345 |
| Prisma | SIEMENS | ADNI3 | 69 | – |
| Prisma Fit | SIEMENS | ADNI3 | 167 | – |

(a)

| Modality | Sequence | Tracers | Dataset | # Coronal slices |
|---|---|---|---|---|
| MRI | FLAIR | N/A | ADNI3 | 526 |
| MRI | T1-weighted | N/A | ADNI3 | 527 |
| MRI | T2-weighted | N/A | ADNI4 | 535 |
| PET | Amyloid | AV45 - FBB - NAV4694 | ADNI3/4 | 516 |
| PET | Tau | AV1451 - MK6240 - PI2620 | ADNI3/4 | 518 |
| PET | FDG | 18F-FDG | ADNI3 | 522 |

(b)

| Modality | Sequence | Tracer | Dataset | # Coronal slices (AD) | # Coronal slices (CN) |
|---|---|---|---|---|---|
| PET | Tau | AV1451 | ADNI3/4 | 530 | 631 |

(c)

(TrioTim, 41 slices for classification, 315 for generation), IXI (Gyroscan Intera: 322/349, Intera: 185/–, GE unspecified: 74/310), and SALD (TrioTim: 494/345).

Modality-conditioned experiments used multimodal MRI and PET data from ADNI3/4 (Jack Jr et al., 2008). Central coronal slices along with ±5 neighboring slices were extracted from MRI (FLAIR, T1, T2) and PET (FDG, amyloid, tau) scans. Slice counts per class were: MRI-FLAIR (526), MRI-T1 (527), MRI-T2 (535), PET-Amyloid (516), PET-Tau (518), and PET-FDG (522), yielding a total of 3,144 slices across all modalities.

### B.2.2 DATASETS FOR CLASSIFICATION

Scanner classification evaluates MAGIC-Flow's ability to discriminate among seven scanner models, including closely related models (e.g., Siemens Prisma vs. Prisma Fit), using unbalanced datasets to test robustness. Coronal slices from PPMI, SALD, IXI, and ADNI3 were used, with the following slice counts: PPMI (TrioTim, 41 slices), IXI (Gyroscan Intera: 322, Intera: 185, GE unspecified: 74), SALD (TrioTim: 494), and ADNI3 (Prisma: 69, Prisma Fit: 167). Data were split into 5 folds, with performance averaged across folds.

Diagnosis classification used Tau PET scans from ADNI3/4 to evaluate MAGIC-Flow's ability to distinguish Alzheimer's disease (AD) from cognitively normal (CN) participants. For each subject, the central coronal slice was extracted. Because the AD group was smaller (106 subjects), training slices were augmented by including ±4 neighboring offsets, resulting in 530 AD slices. CN participants (631 subjects) were not augmented. To prevent data leakage, subjects were split into training,

validation, and test sets before augmentation, ensuring that augmented slices from training subjects did not appear in validation or test sets.

### B.3 EVALUATION METRICS FOR GENERATION AND CLASSIFICATION

We evaluate our method against benchmarking methods using metrics designed for both medical image generation and classification. These metrics capture distributional similarity, sample-level fidelity, diversity, and predictive performance.

To rigorously assess the quality of generated medical images, we employ metrics that evaluate both distributional similarity, which measures how closely generated images match real data, and sample-level fidelity and diversity, which measure how realistic and varied the images appear. This dual perspective provides a comprehensive understanding of generative performance in the medical domain.

#### B.3.1 DISTRIBUTIONAL SIMILARITY METRICS.

The Fréchet Inception Distance (FID) (Heusel et al., 2017) is a standard metric for comparing real and generated image distributions. It measures the Fréchet distance between multivariate Gaussian distributions fitted to deep features, typically extracted from the pool3 layer of an ImageNet-pretrained InceptionV3 network. However, this approach is suboptimal for medical imaging due to the domain gap between natural and medical images.

To address this, we adopt domain-specific adaptations. $FID_{Rad}$ leverages features from an InceptionV3 model pretrained on RadImageNet (Mei et al., 2022), a large-scale medical imaging dataset. This improves alignment with clinical image characteristics and has been shown to correlate more strongly with radiological quality and anatomical fidelity (Fernandez et al., 2024). In addition, we employ $FID_{SwAV}$ (Morozov et al., 2021), which replaces Inception features with embeddings from SwAV, a self-supervised model trained with clustering and contrastive objectives (Caron et al., 2020). Unlike ImageNet features, SwAV embeddings are robust to domain shifts and better suited for grayscale medical images, capturing subtle textures and anatomical structures (Woodland et al., 2024).

Despite its popularity, FID is biased when computed on small sample sizes, a common constraint in medical datasets. This arises because Gaussian parameters are poorly estimated under limited data, resulting in high variance and unstable scores. To mitigate this, we also report the Kernel Inception Distance (KID) (Bińkowski et al., 2018), which computes the squared Maximum Mean Discrepancy using polynomial kernels. Unlike FID, KID is an unbiased estimator and therefore more reliable for small datasets. To maintain consistency with domain-specific features, we compute $KID_{Rad}$ using RadImageNet-pretrained embeddings.

Finally, to further stabilize estimates, we include bootstrapped versions of the domain-adapted metrics, denoted $FID^b_{Rad}$ and $KID^b_{Rad}$. These are obtained via repeated random subsampling, and we report the 95% confidence intervals across subsamples. This reduces variance and improves robustness in small-sample settings.

By combining standard FID with domain-adapted, unbiased, and bootstrapped variants, we construct a robust framework for measuring global distributional similarity between real and generated medical images.

#### B.3.2 SAMPLE-LEVEL FIDELITY AND DIVERSITY METRICS.

While distributional metrics capture overall similarity, they do not directly assess sample-level fidelity (how realistic individual images appear) or diversity (how well the generator covers variations in the data). To address this, we employ a complementary set of metrics: Improved Precision and Recall, Density and Coverage (PRDC), and Structural Similarity (MS-SSIM).

Improved *precision (P)* (Kynkäänniemi et al., 2019) measures the fraction of generated samples that lie within the support of real data, estimated via a fixed-radius nearest-neighbor approach in feature space. Higher values (up to 1.0) indicate greater fidelity, meaning generated images are visually indistinguishable from real ones. Conversely, *recall (R)* (Kynkäänniemi et al., 2019) measures the fraction of real images that are covered by at least one generated sample, reflecting diversity. Higher values (up to 1.0) suggest the generator successfully captures variations in the real distribution.

To provide a more granular view, we additionally compute *density (D)* (Naeem et al., 2020), which counts the average number of generated samples within the neighborhood of each real image using $k$-nearest neighbors. High density (up to 1 in the normalized version) indicates that many generated samples cluster near real images, while low density indicates sparse coverage around real images. Complementing this, *coverage (C)* (Naeem et al., 2020) quantifies the fraction of real images that are matched by at least one generated image within a fixed distance. Coverage ranges from 0 to 1: higher values indicate that the generator spans almost all distinct regions of the real distribution, while lower values indicate that some modes are missing entirely, even if the generator produces realistic samples elsewhere. Whereas recall emphasizes broad inclusion, coverage specifically evaluates whether the generator touches all distinct regions of the real distribution. Interpreted together, density and coverage provide a complete view of generation quality: high density with low coverage signals realistic samples concentrated in few regions (mode collapse), while high density with high coverage indicates both realistic and diverse generation.

All PRDC metrics are computed in the feature space of the RadImageNet-pretrained InceptionV3 model, ensuring domain-specific alignment with medical image characteristics.

To directly assess intra-class and inter-class diversity, we use the Structural Similarity Index Measure (SSIM)(Wang et al., 2004), which compares luminance, contrast, and structural information between two images. In particular, we adopt the Multi-Scale SSIM (MS-SSIM)(Wang et al., 2003), which extends SSIM by evaluating image similarity across multiple spatial resolutions, capturing both coarse and fine details. Following prior work (Odena et al., 2017; Dash et al., 2017; Dragan et al., 2023), we compute MS-SSIM scores over 500 randomly selected pairs of generated images, distinguishing between intra-class and inter-class diversity due to our conditional setting. Intra-class diversity (MS-SSIM$^{\text{intra}}$) is computed among samples of the same class. Lower values indicate greater diversity within a class, while higher values suggest mode collapse toward visually similar outputs. Inter-class diversity (MS-SSIM$^{\text{inter}}$) is computed across samples from different classes. Lower values reflect good class separability, whereas higher values may indicate insufficient differentiation between categories.

### B.3.3 CLASSIFICATION METRICS.

For classification models, we report standard metrics including overall Accuracy, Balanced Accuracy, Area Under the ROC Curve (AUC), and macro-averaged Precision, Recall, and F1-score. Accuracy measures the fraction of correctly classified samples, while Balanced Accuracy accounts for class imbalance by averaging recall across all classes. AUC evaluates the model's discriminative ability across decision thresholds. Macro-averaged Precision, Recall, and F1-score summarize per-class performance and provide a comprehensive view of sensitivity and specificity across categories.

### B.4 BENCHMARK MODELS FOR COMPARISON

To evaluate the effectiveness of MAGIC-Flow, we benchmark it against a diverse set of competitive state-of-the-art models spanning adversarial training, diffusion-based generation, flow-based modeling, and latent-variable approaches. These baselines are well-established in the literature and cover a broad methodological spectrum, ensuring a robust and comprehensive comparison.

### B.4.1 GENERATIVE MODELS

- **SNGAN**: Spectral Normalization GAN (Miyato et al., 2018) introduces spectral normalization to the discriminator's weights, enforcing a Lipschitz constraint that stabilizes training. It is a widely adopted conditional GAN known for generating high-fidelity samples with reliable convergence.

- **StyleGAN2-DiffAug-LeCam**: This approach extends StyleGAN2 (Karras et al., 2020) by incorporating DiffAugment (Zhao et al., 2020) and LeCam regularization (Tseng et al., 2021), which together improve training efficiency and generalization on limited data. These augmentations help maintain strong conditional generation performance without requiring large-scale datasets.

- **ADC-GAN**: The Auxiliary Discriminative Classifier GAN (Hou et al., 2022) enhances the BigGAN framework by integrating an auxiliary classifier that simultaneously predicts

class labels and discriminates real versus fake samples via class-specific labeling. This dual role improves intra-class diversity and promotes stable training dynamics compared to traditional conditional GAN variants like AC-GAN (Odena et al., 2017).

- **CVAE**: The Conditional Variational Autoencoder (Sohn et al., 2015) models conditional distributions via latent variables by injecting conditioning information into both the encoder and decoder networks. This probabilistic approach captures multimodal outputs and enables principled maximum likelihood training.

- **DDPM**: Denoising Diffusion Probabilistic Models (Nichol & Dhariwal, 2021; Dhariwal & Nichol, 2021) are non-adversarial generative models that learn to reverse a fixed noising process through iterative denoising. We adopt the conditional DDPM framework with classifier guidance, which leverages a pretrained classifier's gradients during sampling to steer generation towards the desired class, significantly improving conditional sample quality and fidelity.

- **Medfusion**: Medfusion (Müller-Franzes et al., 2023) is a conditional latent DDPM for medical image synthesis, combining a variational autoencoder for compression with a UNet-based diffusion model operating in latent space. The model uses DDIM sampling for efficient generation and demonstrates superior diversity and fidelity compared to GAN-based methods across ophthalmologic, radiologic, and histopathologic imaging modalities, while producing fewer artifacts in generated medical images.

- **TarFlow**: Transformer AutoRegressive Flow (Zhai et al., 2024) implements Normalizing Flows through Transformer-based autoregressive blocks operating on image patches with alternating directions. The model combines Gaussian noise augmentation during training, post-training score-based denoising, and guidance for both conditional and unconditional generation. TarFlow establishes new state-of-the-art results for likelihood estimation on images and demonstrates sample quality comparable to diffusion models.

- **Conditional RealNVP/Glow**: RealNVP and Glow are normalizing flow models that construct invertible transformations mapping images to latent variables with exact, tractable likelihoods. RealNVP relies on affine coupling layers to enable efficient inversion and log-determinant computation. At the same time, Glow extends this architecture with invertible 1×1 convolutions and multi-scale structures that improve expressivity and sample quality. To include a flow-based baseline in our evaluation, we implement a conditional variant of RealNVP/Glow (Dinh et al., 2016; Kingma & Dhariwal, 2018), as no publicly available conditional Glow models were suitable for our setting.

### B.4.2 Discriminative Models for Classification

To establish a robust comparison framework, we further benchmark against convolutional neural networks (CNNs) pretrained on RadImageNet, as well as Vision Transformers (ViTs).

**CNNs.** RadImageNet is a large-scale medical imaging database containing approximately 1.35 million annotated CT, MRI, and ultrasound images across 11 anatomic regions and 165 pathologic labels (Mei et al., 2022). Unlike ImageNet pretraining on natural images, RadImageNet provides domain-specific initialization that has been shown to improve transferability to radiologic tasks. We selected four widely used CNN architectures, all initialized with RadImageNet-pretrained weights and subsequently fine-tuned on our dataset:

- **ResNet-50** (He et al., 2016) – A residual network that mitigates vanishing gradients with skip connections, providing a strong balance of depth and efficiency.

- **DenseNet-121** (Huang et al., 2017) – A densely connected network that enhances feature reuse and parameter efficiency.

- **InceptionV3** (Szegedy et al., 2016) – An architecture employing factorized convolutions and dimensionality reduction to capture multi-scale features efficiently.

- **InceptionResNetV2** (Szegedy et al., 2017) – A hybrid combining Inception modules with residual connections for deeper feature extraction and stable optimization.

**Vision Transformers.** Unlike CNNs, which learn spatial hierarchies through convolutions, ViTs partition images into fixed-size patches that are linearly projected and processed by transformer encoder layers with self-attention (Dosovitskiy et al., 2020). Prior work has demonstrated that ViTs

pretrained on large datasets (e.g., ImageNet-21k) can achieve strong performance in medical imaging tasks, particularly when combined with CNN backbones (Dosovitskiy et al., 2020; Jain et al., 2024). We include the following variants:

- **ViT-v1/32** – A baseline transformer dividing $224 \times 224$ images into $32 \times 32$ patches.
- **ViT-ResNet/16** – A hybrid model that extracts features with a ResNet backbone before transformer encoding. It uses $16 \times 16$ patches and ImageNet-21k pretraining, enabling improved initialization and faster convergence relative to standalone ViTs.
- **Swin Transformer** (Liu et al., 2021) – A hierarchical vision transformer that introduces shifted window attention to efficiently model both local and global dependencies. Swin achieves state-of-the-art performance on natural image benchmarks such as ImageNet, COCO, and ADE20K, and has been shown to transfer well to medical imaging tasks by capturing fine-grained features while maintaining scalability (Liu et al., 2021; He et al., 2023).

Together, these CNN and ViT baselines provide strong and diverse comparators for evaluating the performance of MAGIC-Flow in medical image classification.

## C   SUPPLEMENTARY DETAILS ON RESULTS

### C.1   DETAILED EFFICIENCY COMPARISON OF GENERATIVE MODELS

Table 5 reports a comprehensive comparison of generative model efficiency measured on an A100 GPU with a batch size of 8. Metrics include inference speed (images per second), total training steps, overall training time, GPU memory usage, and time per training step.

Table 5: Comparison of generative model efficiency on A100 (batch size = 8).

| Model | Inference (img/s) | Training steps | Time on A100 (h) | GPU Memory | Time/step (ms) |
|---|---|---|---|---|---|
| MAGIC-Flow | 37.5 | 74k | 21.9 | 28GB | 1056 |
| TarFlow | 0.11 | 196k | 8.3 | 6GB | 152 |
| Glow | 300 | 94k | 16.9 | 23GB | 645 |
| SNGAN | 49.75 | 100k | 7.15 | 10GB | 257 |
| StyleGAN2 | 52.2 | 100k | 22.8 | 6GB | 821 |
| ADC-GAN | 38.17 | 100k | 26.1 | 21GB | 940 |
| CVAE | 369 | 38k | 1.23 | 6GB | 118 |
| DDPM | 0.24 | 130k | 68.1 | 15GB | 1886 |
| MedFusion | 1.77 | 382k | 32.2 | 6GB | 303 |

### C.2   QUALITATIVE INSPECTION OF THE GENERATED IMAGES

Qualitative results in Figure 7 are consistent with the quantitative trends. MAGIC-Flow produces sharp, anatomically coherent images that preserve scanner- and modality-specific characteristics. In contrast, CVAE outputs appear overly smooth, GAN models exhibit varying degrees of mode collapse, and DDPM samples—while sometimes diverse—often contain noise and reduced anatomical plausibility. MedFusion generally achieves high visual quality, but occasional artefacts from its latent diffusion autoencoder limit the fidelity of fine structural details.

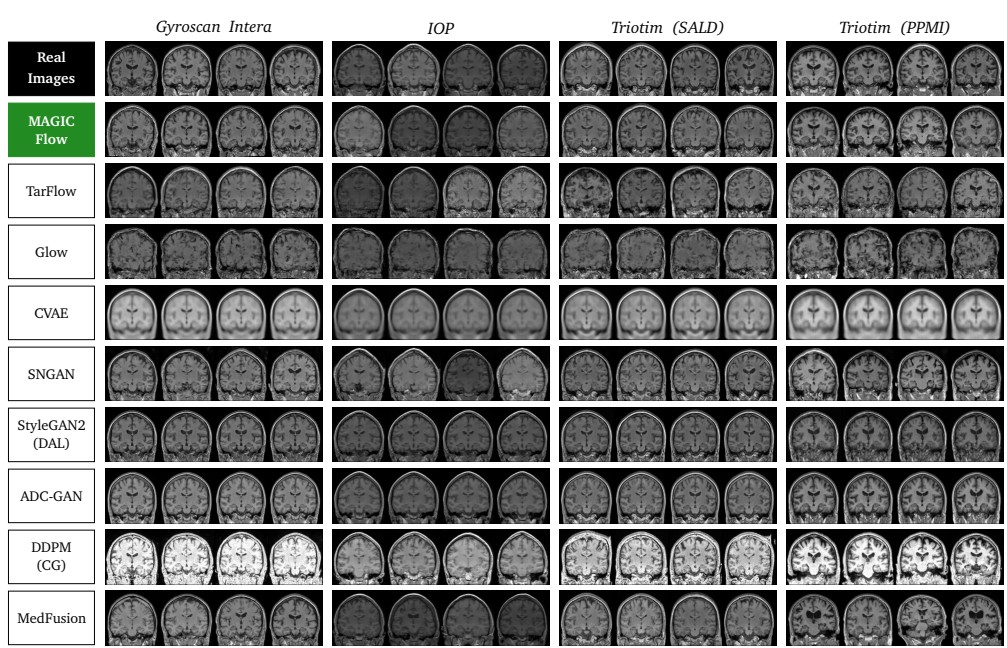

(a) Generated samples across scanner types.

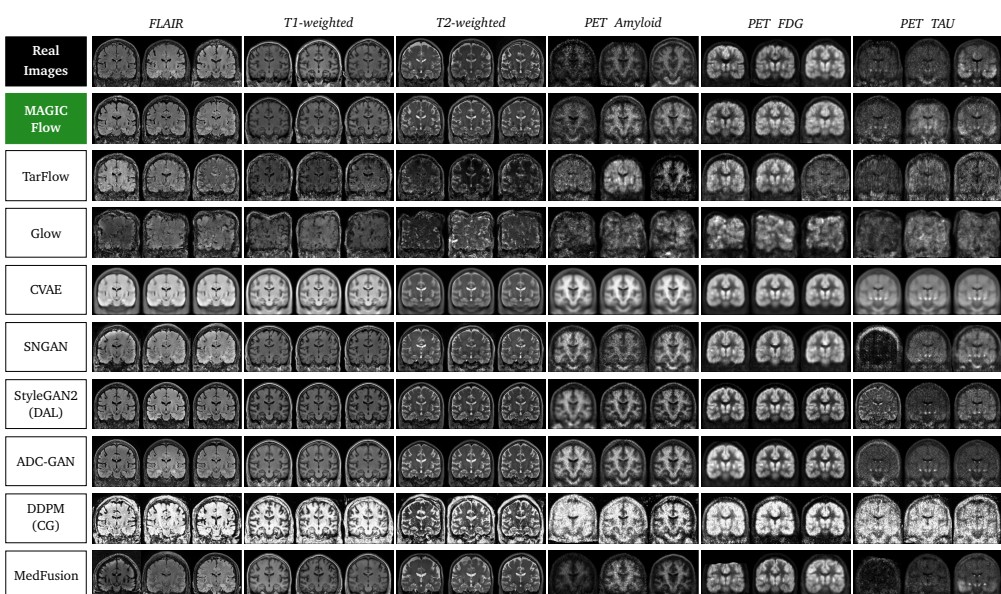

(b) Generated samples across imaging modalities.

Figure 7: Qualitative comparison of generated neuroimaging slices across (a) scanner types and (b) imaging modalities.

