# OpenReview forum: "MAGIC-Flow: Multiscale Adaptive Conditional Flows for Generation and Interpretable Classification"
_ICLR.cc/2026/Conference — Submitted to ICLR 2026_

### Official Review · Reviewer_yfGQ · 2025-10-16

**Soundness:** 3
**Presentation:** 2
**Contribution:** 2
**Rating:** 2
**Confidence:** 4

**Summary:**

The paper proposes MAGIC-Flow, a conditional multiscale normalizing flow framework that unifies image generation and classification within a single invertible architecture.
The model is designed as a hierarchy of bijective transformations with factored Jacobians, allowing exact likelihood computation and stable training.
By leveraging conditional coupling layers, the method supports controllable sample synthesis and likelihood-based classification, and enables explicit visualization of sample likelihoods as an intrinsic interpretability mechanism.
Experiments on medical imaging datasets evaluate both scanner-conditioned and modality-conditioned generation, as well as scanner classification, showing visually coherent and diverse samples with stable performance.

**Strengths:**

1. The paper proposes a normalizing flow model that unifies image generation and classification within a single invertible architecture, demonstrating flexibility across tasks.

2. The use of likelihood attribution maps offers a theoretically grounded, faithful interpretability mechanism that is intrinsic to the model, rather than relying on gradient approximations.

3. The model achieves stable training and competitive quantitative results on scanner- and modality-conditioned generation tasks compared with classic GAN-based methods

**Weaknesses:**

1. The paper mainly compares with GAN-based methods and lacks comparison with more recent state-of-the-art approaches, such as 3D medical latent diffusion models.

2. Evaluation should be extended to more recent and representative datasets (e.g., BraTS2021) to better demonstrate generalizability.

3. The paper presents qualitative visualization of likelihood attribution maps, which are common in flow-matching works; however, further clarification is needed on how these maps relate to classification or generation, and in what way they provide superior interpretability.

4. The scanner-classification experiment lacks clear practical motivation—additional evidence is needed to show how it benefits downstream medical tasks, such as segmentation or diagnosis.

**Questions:**

1. How do the highlighted regions in the likelihood attribution maps quantitatively relate to the model’s decision process in classification or generation?

2. What is the motivation and practical utility of the scanner-classification experiment beyond demonstrating the model’s discriminative capability?

3. Could the proposed likelihood-based explanation and conditional generation framework be extended to dense prediction settings, for instance by conditioning on segmentation masks as structured spatial inputs?

---

> ### Author Response · Authors · 2025-11-21
> **1. Missing Comparisons Against Diffusion Models**
>
> We thank the reviewer for pointing out recent 3D latent diffusion models. Our method, however, operates entirely in the 2D domain and is **designed for high-resolution 2D medical images with limited annotated data**. For this reason, we compare against 2D diffusion models, which constitute the fairest and most directly comparable baseline. Direct comparison with 3D diffusion models would not be appropriate, as they address a **different problem setting**, operate on volumetric representations, and rely on architectural assumptions that do not transfer to our 2D framework. Moreover, adapting 3D diffusion architectures to our high-resolution 2D setting would require substantial redesign and would not reflect their intended use.
> That said, we would be happy to include any specific 2D diffusion models for medical imaging that the reviewer considers relevant, and we welcome suggestions in this direction.
>
> To strengthen our evaluation and provide a fair same-family benchmark, we have considered several state-of-the-art conditional-flow models: MARFlow (unconditional, cannot provide class-conditional likelihoods), Flow++ (attention and variational dequantization; scaling to 200×200 degrades performance), Glow (closest in design, but no conditional likelihood-based implementation available), CaGlow (low-resolution attribute-guided manipulation; training code unavailable), and CaFlow (attribute-encoder-based facial translation at low resolution; not designed for class-conditional synthesis; code unavailable).
>
> Due to memory requirements, architectural mismatches, and scope differences, direct comparison with these models is often infeasible. We therefore evaluate two feasible **conditional-flow baselines** under our resolution and computation constraints:
> - **TARFlow**: employs classifier-free guidance and a hybrid Transformer–flow backbone, without multiscale factorization.
> - **Conditional Glow/RealNVP variant (custom)**: Conditional affine-coupling model with invertible 1×1 convolutions and multiscale decomposition, trained under the same compute budget for a direct same-family comparison.
>
> The updated manuscript will include all feasible conditional-flow baselines evaluated on the same generative tasks, reporting the full suite of metrics (FID, KID, PRDC, MS-SSIM).

---

> > ### Author Response · Authors · 2025-11-21
> > **2. Need for Evaluation on More Representative Datasets (e.g., BraTS2021)**
> >
> > We appreciate the reviewer’s suggestion to incorporate more recent datasets such as BraTS2021. However, our study already uses the most current clinically curated releases of the Alzheimer’s Disease Neuroimaging Initiative (ADNI), specifically ADNI3 (2016–2022) and the ongoing ADNI4 (2022–present). These phases provide some of the most up-to-date and standardized neuroimaging data available. ADNI4, in particular, represents the newest phase of the project, with recruitment and imaging updates continuing through 2023–2024, making it a contemporary and clinically representative testbed for evaluating generation, conditioning, and interpretability. Also our additional AD vs. CN diagnostic evaluation analysis is conducted on both ADNI3 and ADNI4, further demonstrating that our method is validated on modern and clinically relevant datasets.
> >
> > We also recognize the reviewer’s point regarding **dense prediction settings**. Indeed, the proposed likelihood-based explanation and conditional generation framework could be extended to tasks such as **segmentation-guided generation**, where the model conditions on **structured spatial inputs like segmentation masks**. This would allow MAGIC-Flow to synthesize or manipulate images while respecting complex spatial constraints, making it applicable to tumor modeling, lesion synthesis, or other voxel-level clinical applications. Extending MAGIC-Flow to **pathology-focused tasks**—including tumor synthesis, segmentation-conditioned generation, or multimodal tumor modeling on BraTS-like datasets—is therefore a promising direction. We will explicitly highlight this as **future work**, combining both dataset expansion and dense prediction capabilities to broaden clinical applicability.

---

> > > ### Author Response · Authors · 2025-11-21
> > > **3. Need for Quantitative Evaluation of Likelihood Attribution Maps**
> > >
> > > In response to the question on likelihood evaluation, we clarify that the likelihood map directly reflects how frequently specific input patterns are encountered by the model during training. Because regions that the model observes less frequently correspond to lower-probability events—i.e., the blue (more negative) areas of the likelihood map—the map can be used as an effective tool for anomaly detection and for discriminating between potential classification categories. Moreover, these low-likelihood regions provide an intrinsic form of explainability, revealing which inputs lie outside the model’s typical attention patterns. Conversely, the red, high-likelihood areas denote features that the model encounters more often, both during generation and classification, thus offering the complementary perspective to the information conveyed by the low-likelihood regions.
> > >
> > > We added an additional diagnosis prediction task, i.e., classifying between Alzheimer’s disease (AD) vs. Control Normal (CN), using Tau PET. This allowed us to implement a rigorous quantitative evaluation of our likelihood attribution maps in a clinically-interesting scenario that unveil the true power of the likelihood maps. To assess how well the highlighted regions align with disease-relevant tau-aggregate regions usually appearing in AD patients- and thus how they relate to classification or generation tasks -, we perform the following analysis:
> > > - We performed a **thresholding saliency maps** at multiple threshold levels to obtain **tau-related regions of interest (ROIs)**, keeping a different region (the higher the threshold the bigger the region of interest kept).
> > > - We performed the **Extraction of gray-level distribution features** from each ROI.
> > > - We **Trained logistic regression classifiers** on these features, one per each threshold value.
> > > - We **Measured performance as a function of the retained area**, as to highlight how informative is the area kept by the thresholding of the likelihood attribution map.
> > >
> > > We observe that classification performance peaks at **intermediate thresholds**, which retain the disease-associated tracer-uptake regions while excluding irrelevant areas. The table below summarizes the results of this analysis, showing how the F1-score and the proportion of retained area. Specifically, too extended regions (lower threshold) keep both signal and noise; too reduced regions (higher threshold) discard also relevant information; and in-between thresholds highlight regions containing the true informative content, as highlighted from the performance at 0.1 and 0.2 thresholds. Of note, the largest retained region corresponds to the entire brain of the patient, excluding black borders (its normalized area thus corresponds to 0.72 in the table below).
> > >
> > > \begin{array}{| c | c | c |}
> > > \text{\textbf{Threshold}} & \text{\textbf{F1-score}} & \text{\textbf{Proportion of area}} \newline
> > > \hline
> > > 0.0 & 0.87 & 0.72 \newline
> > > 0.1 & 0.88 & 0.57 \newline
> > > 0.2 & 0.88 & 0.35 \newline
> > > 0.3 & 0.82 & 0.19 \newline
> > > 0.4 & 0.78 & 0.09 \newline
> > > 0.5 & 0.71 & 0.045
> > > \end{array}
> > >
> > > These results demonstrate that MAGIC-Flow’s likelihood attribution maps capture **tau-relevant regions** efficiently: thresholds that are too low include irrelevant areas, while thresholds that are too high discard meaningful tau aggregates, reducing classification performance. The performance-to-area ratio highlights that the intermediate thresholds capture the maximal discriminative signal relative to the area considered.

---

> > > > ### Author Response · Authors · 2025-11-21
> > > > **4. Need for Clinically Relevant Downstream Tasks (Diagnosis or Segmentation)**
> > > >
> > > > We agree that scanner identification is not itself a clinically meaningful endpoint. However, it has important methodological value. First, it quantifies the degree of domain shift across scanners, providing a direct measure of how heterogeneous the imaging data are. Second, it serves as a diagnostic test of the learned representation: if the model strongly encodes scanner-specific features, this indicates a potential generalization risk. Additionally, the results inform whether harmonization or domain-adaptation strategies are needed to ensure robustness and fairness across sites and could benefit possible harmonization models. To address this, as anticipated at the previous point, we are adding a **clinically relevant downstream task**: Alzheimer’s disease (AD) vs. cognitively normal (CN) classification using ADNI3/4 Tau PET scans. This setup enables us to directly evaluate how MAGIC-Flow performs on a pathology-level task with true clinical relevance. For this task, we report standard diagnostic metrics—including accuracy, balanced accuracy, F1-score, precision, recall, and AUROC—and we provide corresponding likelihood-based saliency maps to assess interpretability.

---

> > ### Comment · Reviewer_yfGQ · 2025-11-23
> >
> > Thank you for clarifying the 2D nature of the method. My remaining concern is that the evaluation still relies on relatively dated baselines, with the most recent model from 2022 and only the earliest diffusion model included. To better reflect recent progress in 2D medical image generation, it would be helpful to incorporate at least one latest diffusion-based approach, such as a Latent Diffusion variant.
> >
> > I also note that the rebuttal mentions adding TARFlow as a baseline, but this comparison does not appear in the current manuscript. Including results of TARFlow or latest diffusion-based model in the revised manuscript would substantially strengthen the empirical evaluation.

---

> > > ### Author Response · Authors · 2025-11-25
> > >
> > > Thank you for the helpful suggestions. We agree that incorporating a more recent diffusion-based baseline would strengthen the evaluation. In the revised version, we will include a benchmark using MedFusion (repo: https://github.com/mueller-franzes/medfusion; paper: https://arxiv.org/abs/2212.07501). We believe that this choice could meet your advice.
> > >
> > > Regarding TARFlow and the other additions mentioned in the rebuttal: the reason these results do not yet appear in the currently uploaded manuscript is that we are in the process of implementing all revisions and improvements based on the reviewers’ feedback. The updated version of the paper will include the full set of new experiments, including TARFlow, the implementation of Glow and the MedFusion benchmark.

---

### Official Review · Reviewer_DT2P · 2025-10-31

**Soundness:** 3
**Presentation:** 3
**Contribution:** 3
**Rating:** 2
**Confidence:** 4

**Summary:**

The paper introduces MAGIC-Flow, a conditional multiscale normalizing flow that unifies generation and classification within a single invertible backbone. By leveraging conditional affine couplings, squeeze/split hierarchy, and mask scheduling, the model preserves exact likelihoods and stable optimization while switching between tasks through task-specific coupling networks (expressive FiLM/attention blocks for generation vs. streamlined discriminative couplings for classification). Experiments on scanner-conditioned and modality-conditioned medical image generation and on scanner identification show improved FID/KID, PRDC, MS-SSIM, and class-balanced classification metrics over GANs, diffusion, VAEs, and strong CNN/ViT baselines.

**Strengths:**

1.	Unified and practical design: The method uses the same invertible network for both generation and classification, leveraging exact likelihood computation for both. This is a conceptually clean and theoretically sound approach.
2.	The method's performance is demonstrated on a challenging, domain-specific problem in medical imaging.

**Weaknesses:**

1. Novelty concerns: “Unified generation + classification” sounds incremental relative to prior conditional flows. The paper claims the first conditional multiscale flow that supports both generation and classification on a shared invertible backbone. Yet the Related Work already notes conditional normalizing flows (cNFs), cINNs, and CAFLOW as frameworks for modeling p(x|y)and using label-conditioned transformations, which naturally enable likelihood-based decisions. The manuscript does not clearly articulate a necessary capability gap in cINN/CAFLOW/cNF that MAGIC-Flow uniquely closes.
2. Missing same-family baselines (cINN/CAFLOW/cNF): Results compare against GAN/DDPM/CVAE for generation and CNN/ViT for classification, but not against strong conditional-flow baselines under matched parameters and budgets.
3. Computational Complexity and Scalability: Normalizing flows are notoriously computationally expensive and memory-intensive due to the requirement of calculating Jacobian determinants. The paper's multiscale architecture with 24 flow steps, while expressive, likely exacerbates this issue. A discussion on training/inference time, memory footprint, and scalability to higher resolutions (e.g., 3D volumes, which is noted as a limitation) compared to baselines is missing but crucial for assessing practical utility.
4. Limited Analysis of the "Unified" Claim: While the framework is unified, the implementation requires two different, task-specific coupling networks (for generation and classification). The paper does not adequately explore the interchangeability or shared learning between these two "modes." An ablation studying a single, general-purpose coupling layer for both tasks would have strengthened the claim of a truly unified model.
5. Indirect Classification Mechanism: The classification is performed by comparing likelihoods p(x∣y) for all possible y, which requires running the forward pass once for each class. This is an O(C) operation at inference time, making it significantly less efficient than a single-pass discriminative model (O(1)) when the number of classes C is large. This scalability limitation for large-class problems is not discussed.
6. Niche Applicability of Classification Task: The primary classification task demonstrated is "scanner identification," which, while challenging, is more of a technical or calibration problem than a core clinical task (e.g., disease diagnosis). The paper would be stronger if it also included results on a canonical pathology classification task to demonstrate the generalizability of its discriminative capabilities to semantically meaningful categories.

**Questions:**

Although the method emphasizes minimal changes across tasks, the generation and classification couplings embed numerous capacity-boosting modules—FiLM conditioning, CBAM attention, residual blocks, global context, and ASPP/SE—introduced at multiple depths. This looks like substantial architectural augmentation rather than a light tweak, and the paper lacks component-wise ablations to justify necessity vs. over-engineering. A clear complexity–benefit tradeoff is needed.

---

> ### Author Response · Authors · 2025-11-20
> **1. Novelty and Positioning Relative to Prior Conditional Flows**
>
> We thank the reviewer for the thorough reading and detailed comments. We appreciate the acknowledgment of the conceptual clarity and practical design of MAGIC-Flow, as well as its application to the challenging domain of medical imaging. Below, we address the specific concerns.
>
> We agree that prior conditional flows (cINN, cNF, CaFlow) provide mechanisms for modeling p(x∣y). However, MAGIC-Flow extends beyond that and addresses a concrete capability gap in current literature - of flows and beyond - regarding high-resolution, small-sample medical imaging tasks as well as other tasks with similar constraints. Specifically, we provide:
> - **Unified, invertible backbone for both generation and classification**: While cINN and cNF allow label-conditioned likelihoods, they are typically not designed to simultaneously: (1) preserve multi-scale structure at high resolutions (~200×200); (2) provide stable likelihoods for both generative and discriminative inference; and (3) enable tractable interpretability via explicit likelihood maps.Adapting existing flows to these settings often leads to instability or requires memory-intensive simplifications that reduce performance. We will deepen the discussion of prior work in this directions, stressing the actual advantages of the proposed approach.
> - **Domain-specific design**: Medical imaging imposes stringent constraints, including limited labeled data, subtle pathological differences, high-resolution inputs, and the need for clinically meaningful interpretability. MAGIC-Flow’s architecture—including multi-scale factorization, invertible conditioning, and task-specialized couplings—directly addresses these challenges, specifically allowing inherent interpretability and showing stability even in settings with small batch size.

---

> ### Author Response · Authors · 2025-11-20
> **2. Missing Same-Family Baselines**
>
> We fully agree that including same-family conditional-flow baselines is important for a fair assessment of MAGIC-Flow. However, **practical constraints** make direct comparisons non-trivial:
> - **Lack of available code**: Most state-of-the-art conditional flows do not release training code or only provide low-resolution examples. Re-implementing them introduces performance variability, making a fair benchmark difficult.
> - **High-resolution medical images (≈200×200)**: Existing models are optimized for 32–64 px datasets. Scaling to high resolution drastically increases memory usage (up to >80GB for exact likelihood training) and often requires simplifying attention, coupling, or multiscale mechanisms, which compromises performance.
> - **Architectural incompatibility with classification**: Many conditional flows support generation but cannot be directly adapted to likelihood-based classification without major redesigns—e.g., recalibrating log-likelihoods for discriminative inference, modifying conditioning paths, or re-architecting multiscale couplings.
> - **Scope mismatch between generation and classification**: While candidate flows may be able to generate high-fidelity images, adapting them for classification in a stable and interpretable way is not trivial and effectively constitutes new model development.
>
> We have considered several state-of-the-art conditional-flow models:
> - **MARFlow**: Unconditional; cannot provide class-conditional likelihoods.
> - **Flow++**: Variational dequantization and attention; originally designed for low-resolution natural images, scaling to 200×200 requires simplifications that degrade performance.
> - **Glow**: Most similar in design, but no publicly available conditional Glow implementation supports likelihood-based classification with class labels.
> - **CaGlow**: Designed for low-resolution attribute-guided image manipulation; training code unavailable.
> - **CaFlow**: Uses attribute encoders and cycle-consistency for facial translation at low resolution; not designed for class-conditional synthesis, training code unavailable.
> Due to these practical constraints—including high memory requirements for 200×200 images, architectural incompatibility, and scope mismatch—direct comparison with these models is not feasible.
>
> To address this, we are evaluating two feasible conditional-flow baselines under our resolution and compute constraints:
> - **TARFlow**: employs classifier-free guidance and a hybrid Transformer–flow backbone, without multiscale factorization.
> - **Conditional RealNVP/Glow variant (custom implementation)**: to ensure a same-family comparison, we are implementing a conditional affine-coupling model with invertible 1×1 convolutions, with multiscale decomposition, trained under the same compute budget.
>
> In the updated manuscript, we will include all feasible conditional-flow baselines that train robustly at 200×200 resolution within our GPU memory limits. We will evaluate them on the generative tasks presented in the paper and report the full suite of metrics (FID, KID, PRDC, MS-SSIM). These additions will enable a fair and transparent comparison and further demonstrate how MAGIC-Flow’s design supports strong performance in generative settings for high-resolution, small-sample medical imaging. Should the reviewer have other suggestions for the implementation of and the comparison with baseline models, we wil include them in the analysis.

---

> ### Author Response · Authors · 2025-11-20
> **3. Computational Complexity, Memory, and Scalability**
>
> Below we provide a detailed comparison of training and inference runtime, as well as GPU memory usage, using a reference batch size of 8 for all models and an A100 40 GB GPU. Although MAGIC-Flow incorporates a multi-scale architecture to increase expressiveness, it is deliberately engineered to remain stable even with very small batch sizes (e.g., batch size 2) and to avoid unnecessary memory growth. While classification via likelihood comparison scales as O(C), this cost remains practical in medical imaging settings, where the number of clinically meaningful classes is typically small.
> \begin{array}{| c | c | c |  c | c |}
> \text{\textbf{Model}} & \text{\textbf{Inference (img/s)}} & \text{\textbf{Training steps}} & \text{\textbf{Time on A100}} & \text{\textbf{GPU Memory}} \newline
> \hline
> \text{MAGIC-Flow} & 37.50 & 74,670 & 21.9\ h & 28\ GB \newline
> \text{SNGAN} & 49.75 & 100,000 & 7.15\ h & 10\ GB \newline
> \text{StyleGAN2} & 52.20 & 100,000 & 22.8\ h & 6\ GB \newline
> \text{ADC-GAN} & 38.17 & 100,000 & 26.1\ h & 21\ GB \newline
> \text{CVAE} & 369.00 & 37,500 & 1.23\ h & 6\ GB \newline
> \text{DDPM} & 0.24 & 130,000 & 68.1\ h & 15\ GB \newline
> \text{TarFlow} & 0.11 & 196,500 & 8.3\ h & 6\ GB
> \end{array}
> We further report a scaling analysis of inference time as a function of image resolution, which is particularly relevant for clinical deployment scenarios where images can vary substantially across modalities. The analysis was conducted with an A100 80GB GPU.
> \begin{array}{| c | c |}
> \text{\textbf{Image Size}} & \text{\textbf{Generation Time (imgs/s)}} \newline
> \hline
> 1 \times 182 \times 182 & 42.33 \newline
> 1 \times 224 \times 224 & 30.45 \newline
> 3 \times 256 \times 256 & 23.47 \newline
> 3 \times 384 \times 384 & 10.59 \newline
> 3 \times 512 \times 512 & 5.97
> \end{array}

---

> > ### Author Response · Authors · 2025-11-20
> > **4. Analysis of Unified vs. Task-Specific Couplings**
> >
> > We agree that our couplings include expressive modules for generation and streamlined modules for classification. In principle, classification would still work correctly using the more expressive, generation-oriented coupling. However, ablation studies demonstrate that classification performance is maintained with the lightweight discriminative couplings, indicating that the additional capacity in the generator couplings is unnecessary for classification. Thus, the expressive generator coupling is over-parameterized for classification. Using a shared, heavy coupling for both tasks does not improve discriminative performance, but substantially increases computational cost and memory usage. We will include the above-mentioned ablation study and complexity–benefit analysis in the revision to clarify this design choice. Should the reviewer have specific suggestion or constraints for improving such ablation study we will be glad to discuss.

---

> ### Author Response · Authors · 2025-11-20
> **5. Indirect Classification Mechanism and O(C) Inference**
>
> We acknowledge that likelihood-based classification requires an O(C) forward pass. While this is less efficient than a single-pass discriminative model, it **provides exact class-conditional probabilities** and integrates naturally with the generative pathway. For medical imaging, the number of clinically relevant classes is often small (2–10), making this a reasonable trade-off. We will clarify these considerations and discuss practical strategies (e.g., caching intermediate activations or hierarchical classification) to reduce inference cost when C is larger. If needed, we can also report inference-time comparisons against standard discriminative baselines to quantify the computational overhead in the classification setting.

---

> ### Author Response · Authors · 2025-11-20
> **6. Applicative Relevance: Canonical Pathology Classification**
>
> We recognize that scanner identification is a technical task. To demonstrate generalizability to semantically meaningful tasks, we additionally perform the task on Tau PET from ADNI3/4, using a canonical pathology-level task: Alzheimer’s disease (AD) vs. cognitively normal (CN) classification. We report standard performance metrics—including accuracy, balanced accuracy, F1-score, recall, precision, and AUROC—along with saliency maps for interpretability. To quantitatively assess the informativeness of these maps, we additionally perform a logistic regression analysis thresholded ROI–feature as described below, as requested by other reviewers.
>
> **Quantitative Interpretability Assessment**
>
> The added AD vs. CN task also enables a rigorous quantitative evaluation of our likelihood attribution maps. To assess alignment with disease-relevant tau regions, we threshold the saliency maps at multiple levels to define tau-related ROIs, extract gray-level features from each ROI, and train logistic regression classifiers. Performance is discussed as a function of retained area. We find that classification peaks at intermediate thresholds, which retain disease-associated tracer uptake while excluding irrelevant regions. The table below summarizes the results, showing F1-score and proportion of retained area across thresholds, highlighting that intermediate thresholds capture maximal discriminative signal relative to area. The largest retained region corresponds to the entire brain of the patient, excluding black borders (its normalized area corresponds to 0.72 in the table below).
>
> \begin{array}{| c | c | c |}
> \text{\textbf{Threshold}} & \text{\textbf{F1-score}} & \text{\textbf{Proportion of area}} \newline
> \hline
> 0.0 & 0.87 & 0.72 \newline
> 0.1 & 0.88 & 0.57 \newline
> 0.2 & 0.88 & 0.35 \newline
> 0.3 & 0.82 & 0.19 \newline
> 0.4 & 0.78 & 0.09 \newline
> 0.5 & 0.71 & 0.045
> \end{array}
>
> These analyses show that MAGIC-Flow highlights clinically meaningful regions, supporting both its generative fidelity and its discriminative interpretability.

---

### Official Review · Reviewer_GBzt · 2025-11-01

**Soundness:** 3
**Presentation:** 4
**Contribution:** 3
**Rating:** 6
**Confidence:** 4

**Summary:**

The paper introduces MAGIC-Flow, a conditional multiscale normalizing flow designed to unify image generation and classification within a single invertible framework. Unlike standard flows (e.g., RealNVP, Glow) that are purely generative, MAGIC-Flow leverages conditional  mappings to perform both tasks with exact likelihood estimation and explicit interpretability.

The architecture integrates hierarchical flow steps, squeeze and split operations, and task-specific  layers to capture multi-scale structure efficiently. Two coupling variants are proposed: one for expressive, label-conditioned generation (using FiLM  CBAM attention), and one for discriminative classification via likelihood-based decision-making.

Experiments on multiple medical imaging datasets (MRI, PET) demonstrate strong performance on both conditional generation and classification . MAGIC-Flow achieves substantially lower FID/KID scores than GAN, diffusion, and VAE baselines, while matching or exceeding CNN and ViT classifiers in accuracy and balanced recall.

**Strengths:**

I find this paper to be technically strong and clearly presented. The authors address an important gap between generative and discriminative modeling in medical imaging by proposing an invertible architecture that unifies both within a single framework. This is an elegant idea, as existing flow-based methods typically require separate models or auxiliary classifiers.

The paper provides a solid theoretical foundation showing that Jacobian factorization extend naturally to the conditional setting.  I also appreciate the task-specific layers, which are carefully adapted for generation and classification..

The empirical evaluation is extensive and convincing. The authors benchmark across multiple public MRI and PET datasets, demonstrating strong improvements in both generation quality (FID/KID, PRDC metrics) and classification performance (accuracy +  F1). The visual results look realistic and diverse.

**Weaknesses:**

While the paper and the idea of unifying generation and classification in a conditional flow is appealing, I find the degree of novelty to be moderate. The architecture largely builds upon established components from RealNVP and Glow, and while the conditional and task-specific extensions are thoughtfully designed, they remain incremental rather than groundbreaking. The main conceptual contribution is more in integration and adaptation than in introducing fundamentally new flow mechanisms.

From an experimental standpoint, although the results are strong, the evaluation focuses almost entirely on medical imaging datasets (MRI and PET). Furthermore, most comparisons are made against GANs, diffusion, and VAEs, but not against recent state-of-the-art conditional flows.

A second limitation is interpretability validation: while the proposed likelihood attribution maps are interesting and conceptually sound, their evaluation remains qualitative. It would be beneficial to include quantitative validation to demonstrate that these maps meaningfully explain the model’s decisions.

Finally, although the paper discusses clinical trustworthiness and interpretability, there is limited discussion on computational cost and scalability. Flow-based models are often resource-intensive, and it would help to see clearer comparisons of runtime, parameter count, and memory usage against other generative models.

**Questions:**

- Could you clarify how MAGIC-Flow fundamentally differs from prior conditional or multi-scale flow models
- Flow-based models are often resource-heavy. Could you provide more details on runtime, memory , and training stability compared to baselines? How does inference time scale with image size?

---

> ### Author Response · Authors · 2025-11-20
> **1. Novelty and Positioning Relative to Prior Conditional / Multiscale Flow Models**
>
> We appreciate the reviewer’s observation that MAGIC-Flow builds on established flow components. Our core contribution, however, extends beyond the integration of different existing components towards the definition of a new coupling transformation in a novel (to the best of our knowledge) **unified, invertible, multi-task design** that supports
>  - expressive label-conditioned generation,
>  - the extraction and exploitation of class-conditional likelihoods for classification,
>  - multi-scale factorization optimized for high-resolution medical images, and
>  - a discriminative pathway grounded entirely in invertibility rather than auxiliary classifiers. These points are missing from literature of both flows and generative models.
>
> A further crucial aspect of our contribution is that we target a particularly challenging applicative domain—medical imaging—where incremental architectural decisions become essential.
> Medical imaging presents a unique combination of constraints and thus provide the perfect room for improving along the following directions:
> - **Extremely limited labeled data** compared to natural-image benchmarks, which makes overfitting a severe risk and demands models with stable likelihoods and strong inductive biases.
> - **High-resolution inputs (≈200×200)** with complex anatomical structures, which render many natural-image flow architectures unusable without significant modification or prohibitively high memory requirements.
> - **Fine-grained, clinically meaningful differences across classes**, requiring architectures that preserve fine-grained spatial detail and allow targeted uncertainty estimation.
> - **Strict interpretability requirements**, since clinical deployment demands transparent decision mechanisms beyond black-box classification.
>
> MAGIC-Flow is purposely built to address these constraints, currently not handled (at least together) in literature:
> - The **multi-scale decomposition** is tailored to preserve local anatomical structure while maintaining tractable memory usage.
> - The **conditional design** enables both generation and classification from the same invertible backbone, reducing data requirements and improving robustness in small-sample regimes.
> - The **exact likelihoods and invertible transformations** provide explicit, quantitative interpretability, which is essential for medical AI but unavailable in GANs or diffusion models.
> - The architecture remains **stable at extremely small batch sizes**, an important practical requirement when training on high-resolution 3D/2D medical data with limited GPU memory.
>
> To clarify these distinctions, we will expand the Related Work section contrasting MAGIC-Flow with Glow, cINN, CaFlow, Flow++, and MARFlow, while highlighting which components are necessary to satisfy medical-imaging (as well as other fields with similar) constraints.

---

> > ### Author Response · Authors · 2025-11-20
> > **2. Missing Comparisons Against Conditional Flow Baselines**
> >
> > We agree this comparison is valuable and have begun implementing additional baselines. However, several practical constraints limit direct benchmarking:
> > - **Lack of available implementations**: Many state-of-the-art conditional flows do not release training code. Re-implementing them introduces performance discrepancies that make the comparison less reliable.
> > - **Architectural incompatibility with high-resolution 2D medical imaging**: Most flow benchmarks use 32–64 px images. Extending these models to 200×200 exact-likelihood training frequently exceeds 80GB memory (that is the usual available computation resource in clinical settings, as well as ours). This requires architectural simplifications that significantly degrade their reported performance.
> > - **Scope mismatch (generation vs. classification)**: Many conditional flows support conditional generation but not likelihood-based classification. Adapting them would require substantial model redesign, effectively creating new methods. We will clarify this in the manuscript.
> >
> > Candidate baselines considered are:
> > - **TARFlow** employs classifier-free guidance and a hybrid Transformer–flow backbone, but without multiscale factorization.
> > - **Flow++** relies on variational dequantization and invertible attention, which become computationally expensive at our input resolution.
> > - **MARFlow** is unconditional and does not support class-conditional likelihoods.
> > - **Glow** is the most similar approach; however, no conditional Glow with released code supports class-label conditioning or likelihood-based classification.
> > - **CaGlow** extends Glow with an adversarial encoder for low-res attribute-guided image manipulation; training code not publicly available.
> > - **CaFlow** uses an attribute encoder and cycle-consistency losses for controllable facial image translation; designed for low-res editing, not class-conditional synthesis; training code unavailable.
> >
> > To strengthen our experimental comparisons, we will add TARFlow and a custom conditional Glow implementation (featuring affine coupling, invertible 1×1 convolutions, and multiscale decomposition) to the paper. We would appreciate recommendations for other flow-based models that could serve as informative baselines.

---

> ### Author Response · Authors · 2025-11-20
> **3. Interpretability: Need for Quantitative Validation**
>
> We thank the reviewer for highlighting this important aspect. In the revision, we will include a fully quantitative assessment of our likelihood-based saliency maps. Following the suggestion of other reviewers, we introduce a clinically meaningful diagnosis-prediction task—Alzheimer’s disease classification (AD) versus cognitively normal (CN) subjects—using Tau PET data from ADNI3/4. This newly added AD vs. CN task enables a rigorous quantitative evaluation of our likelihood-based attribution maps, in an interesting applicative setting.
> Our procedure is as follows:
> - **Region segmentation via thresholding**: We segment the saliency maps by applying multiple thresholds, producing regions of interest (ROIs) corresponding to the most salient areas highlighted by MAGIC-Flow. The largest retained region corresponds to the entire brain of the patient, excluding black borders (its normalized area corresponds to 0.72 in the table below).
> - **Feature extraction from ROIs**: From each segmented region, we extract features describing the distribution of gray-level intensities, capturing patterns of tracer uptake or tissue activity relevant to disease processes. These features are then used as inputs to logistic regression models for disease classification (CN vs AD).
> - **Threshold-performance analysis**: We observe that classification performance depends critically on the choice of threshold, as highlighted in the Table below:
>
> \begin{array}{| c | c | c |}
> \text{\textbf{Threshold}} & \text{\textbf{F1-score}} & \text{\textbf{Proportion of area}} \newline
> \hline
> 0.0 & 0.87 & 0.72 \newline
> 0.1 & 0.88 & 0.57 \newline
> 0.2 & 0.88 & 0.35 \newline
> 0.3 & 0.82 & 0.19 \newline
> 0.4 & 0.78 & 0.09 \newline
> 0.5 & 0.71 & 0.045
> \end{array}
> These results indicate that using the entire image dilutes the informative signal, resulting in lower classification performance, while selecting an overly restrictive threshold discards relevant information, similarly reducing performance. The optimal threshold aligns with regions highlighting clinically meaningful tracer uptake, demonstrating that MAGIC-Flow’s likelihood-based maps accurately identify disease-relevant structures. We believe that these findings provide strong quantitative evidence that MAGIC-Flow concentrates its probability mass and discriminative signal on pathologically relevant regions, thus acting as a ROI segmentation for additional downstream analyses. We will retain illustrative qualitative examples and expand the Discussion to emphasize how this approach enhances interpretability, reliability, and clinical trustworthiness.

---

> ### Author Response · Authors · 2025-11-20
> **4. Computational Cost, Memory Footprint, and Scalability**
>
> We will provide full tables of GPU memory usage, training/inference runtime across all feasible baselines and ours. All measurements are reported with a reference batch size of 8  and an A100 40 GB GPU for fair comparison.
> A key observation we may point out is that many generative models exhibit substantial variability in training stability. For instance, some GANs and flow-based models can experience performance degradation when memory constraints force small batch sizes, plateau early at sub-optimal performance, or require extended warm-up schedules to stabilize gradients. In contrast, **MAGIC-Flow** demonstrates **stable training** across a wide range of batch sizes, including very small ones (e.g., batch size 2), making it far more adaptable under compute-constrained environments.
> \begin{array}{| c | c | c |  c | c |}
> \text{\textbf{Model}} & \text{\textbf{Inference (img/s)}} & \text{\textbf{Training steps}} & \text{\textbf{Time on A100}} & \text{\textbf{GPU Memory}} \newline
> \hline
> \text{MAGIC-Flow} & 37.50 & 74,670 & 21.9\ h & 28\ GB \newline
> \text{SNGAN} & 49.75 & 100,000 & 7.15\ h & 10\ GB \newline
> \text{StyleGAN2} & 52.20 & 100,000 & 22.8\ h & 6\ GB \newline
> \text{ADC-GAN} & 38.17 & 100,000 & 26.1\ h & 21\ GB \newline
> \text{CVAE} & 369.00 & 37,500 & 1.23\ h & 6\ GB \newline
> \text{DDPM} & 0.24 & 130,000 & 68.1\ h & 15\ GB \newline
> \text{TarFlow} & 0.11 & 196,500 & 8.3\ h & 6\ GB
> \end{array}
> We also present a scaling analysis of inference time versus image size, which is especially relevant for clinical deployment where inference speed—not training time—is the primary bottleneck to account for. The analysis was conducted with an A100 80GB GPU.
> \begin{array}{| c | c |}
> \text{\textbf{Image Size}} & \text{\textbf{Generation Time (imgs/s)}} \newline
> \hline
> 1 \times 182 \times 182 & 42.33 \newline
> 1 \times 224 \times 224 & 30.45 \newline
> 3 \times 256 \times 256 & 23.47 \newline
> 3 \times 384 \times 384 & 10.59 \newline
> 3 \times 512 \times 512 & 5.97
> \end{array}

---

### Meta-Review · Area_Chair_Rhtc · 2025-12-23

**Summary:**

This paper presents a multiscale adaptive conditional flow for generation and classification. While some of the reviewers find the presentation clear and theoretic, it suffers from several major weakness, including, missing comparisons against conditional flow baselines. The author respond that this is duo to many state-of-the-art conditional flows do not release training code while re-implementing them introduces performance discrepancies that make the comparison less reliable. However, I believe this is not a good excuse to skip the comparison. The authors shall instead try to re-implement and report the results. The reviewers also raised the concern on limited to moderated novelty (Reviewer DT2P and Reviewer GBzt). As there are some related Work already notes conditional normalizing flows, the claim of the first conditional multiscale flow that supports both generation and classification seems not convincing. In summary, the paper presents some method theoretical reasonable and clear, but the evaluation is weak.

**Reviewer Concerns:**

The major concerns include limited novelty and insufficient experiments. From the rebuttal, I do not see the authors address this satisfactorily.

**Reviewer Scores:**

Based on the fact that majority of the reviewers do not appreciate the evaluation and novelty of the work, and also the rebuttals provided, I do not think the reviewers would increase their scores significantly.

---

### Decision · Program_Chairs · 2026-01-26

Reject